# Assessing the role of regional educational integration policies in fostering university-industry-research innovation: Evidence from China

Qi Chen, Jiawen Zhou 🔟 *

School of International Economics and Trade, Shanghai Lixin University of Accounting and Finance, Shanghai, China

* 20180083@lixin.edu.cn

## Abstract

Regional integration is a key strategy for enhancing competitiveness and promoting innovation, and regional education integration has become an important part of regional integration. At the same time, university-industry-research (UIR) collaborative innovation is also a key link in building a national innovation system. This study empirically analyzes the impact of China's Yangtze River Delta regional education integration policy on UIR innovation cooperation by using a difference in difference model. The results show that the regional education integration policy significantly promotes UIR innovation cooperation, especially in Zhejiang and Jiangsu, which have a stronger economic base and richer educational resources. In addition, the policy has an extraordinarily prominent role in promoting invention patents with higher innovation intensity. The mechanism analysis finds that human capital, economic capital, government support and social stability play an important mediating role. Finally, policy recommendations are made based on the empirical results.

## 1 Introduction

Innovation has become a crucial factor for economic development, national competitiveness, and sustainable development due to the rapid growth of the global economy and technology. Collaborative innovation between universities, industries, and researchers (UIR) is a critical part of regional innovation systems. It has also become an effective innovation model and an important way to enhance autonomous innovation capacity and drive technological progress across countries and regions [1,2]. However, in practice, fragmented policies, unequal resource distribution, and lack of effective collaboration platforms have confronted UIR collaborative innovation with a series of challenges, which are particularly prevalent in emerging economies [3,4]. It remains an important research question how to effectively promote UIR collaborative innovation through policies, especially in emerging market economies [5].

**Data availability statement:** All relevant data are within the paper and its Supporting information files.

**Funding:** This study was supported by the National Education Sciences Planning of China (Project No. DIA200347), with Qi Chen as the recipient who played an important role in the study design, data collection and analysis, decision to publish, and preparation of the manuscript.

**Competing interests:** The authors have declared that no competing interests exist.

This study is grounded in The Growth Pole Theory, Synergetics Theory, and Innovation Theory, which collectively provide a robust framework for analyzing the role of policies in fostering regional innovation. The Growth Pole Theory emphasizes the role of economic core areas as hubs for innovation and resource concentration, which can drive development in surrounding regions. Studies show that higher education quality is a key to human capital and technological innovation, driving regional economic growth [6]. Zhang [7] applies the theory to explain the relationship between higher education and regional innovation, emphasizing the agglomeration effect in enhancing innovation capacity. Synergetics Theory highlights the importance of coordination among diverse innovation actors—such as universities, industries, and governments—in achieving dynamic equilibrium and fostering collaborative innovation. Jafarov [8] uses this framework to reveal how the collaboration of education and innovation elements optimizes regional development. Wang et al. [9] apply the theory to analyze the interaction between higher education and regional innovation, highlighting its impact on regional economic and innovation growth. Innovation Theory, on the other hand, underlines the mechanisms through which policies can stimulate technological progress and enhance the overall innovation capacity of regions. Li et al. [10] show that education plays a central role in regional innovation, with its interaction with innovation and digitalization aligning with the core concepts of "human capital" and "innovation ecosystems." Qing et al. [11] uses Innovation Theory to explore the impact of regional policies, resource allocation, and human capital on innovation efficiency. These theoretical perspectives are particularly relevant for understanding how regional education integration policies can address structural barriers to innovation and promote UIR collaboration.

Globally, regional education policies have been a key tool for promoting UIR collaborative innovation, especially within the European Union [12] and numerous developed countries [13]. These studies emphasize the importance of regional education policies for optimizing innovation resource allocation and facilitating cross-sector collaboration. Collaborative innovation has become a vital component in advancing national economies, particularly in China, where it is central to transitioning towards a more innovation-driven economy. But studies with precise and quantitative measures of UIR collaboration outcomes are relatively few, and there is a lack of focus on the dynamic interactions among collaborating regions. Moreover, few quantitative studies have examined the mechanisms through which policies are implemented, including how policies perform differently in different economic and social settings [4], or heterogeneity analysis. Additionally, although regional education integration policies have gradually gained global recognition, especially in developed countries, research on their application in emerging economies such as China is still limited, making it difficult to fully apply existing policy experiences to developing countries [14].

In recent years, China has made significant strides in the expansion of its higher education system, increasing educational opportunities and enhancing human capital, particularly in STEM fields. These developments have been closely aligned with China's national innovation goals, contributing to the acceleration of socio-economic and innovation capabilities [15]. The integration of regional education systems plays

a key role in aligning educational advancements with the nation's broader innovation agenda, which focuses on fostering technological advancement, industrial upgrades, and global competitiveness. By improving the quality and availability of education, China has strengthened its economic competitiveness and technological capacity [16–18]. In this context, the implementation of regional education policies, especially those promoting educational integration, has become a crucial driver of innovation and regional economic development, facilitating more effective collaboration between universities, industries, and research institutions.

Unlike the market-driven regional education integration policies of Europe and the United States, China's education policies are typically government-driven, with an emphasis on resource integration, policy guidance, and regional coordination [19]. Specifically, in the Yangtze River Delta (YRD) which is an important region in China with developed economy and industry, high degree of openness and leading the realization of high-quality development in China, the government has promoted the sharing and flow of educational, research, and industrial resources through regional education integration policies, thus facilitating deep university-industry-research cooperation [20,21]. However, China's uneven regional economic development and decentralized education system have led to its increasing regional education disparities [22]. Regions with strong economic foundations and abundant educational resources have been able to implement regional education policies efficiently and achieve significant innovation results. While in economically underdeveloped regions in China, the effects of the policies show significant variation [23]. This regional disparity presents clear heterogeneity in the implementation of China's regional education integration policies.

This study addresses this gap by quantitatively assessing the impact of regional education integration policies on UIR collaborative innovation, focusing on regional disparities and the different levels of innovation intensity across regions. We employ a Difference-in-Differences (DID) model to quantitatively assess the impact of regional education integration policies on UIR collaborative innovation in the Yangtze River Delta region in China. In addition to analyzing the overall effect of the policies, it also investigates the heterogeneous effects across provinces and levels of innovation intensity, exploring the mediating role of endogenous factors such as human capital, government support, economic capital, and social stability. This research provides important theoretical support and policy recommendations for optimizing regional education policies in China and other emerging economies.

## 2 Literature review

### 2.1 Definition of regional UIR collaborative innovation

Regional UIR collaborative innovation refers to collaborations between enterprises, universities, and research institutions within a specific geographic region for the purpose of supporting regional technologic innovation and economic development. The core purpose of regional collaborative innovation lies in the cooperation and resource sharing between various innovation entities to solve technological problems and promote the transfer of technology, which leads to regional economic and industrial growth [24]. In addition to stimulating economic development locally, this innovation model increases the region's competitiveness in the global market, especially in the high-tech industry, where innovation clusters play a critical role [25].

Regional collaborative innovation involves the integration of knowledge and technologies across different industries and is an ongoing evolutionary process [26]. Efficacy of regional collaboration innovation is not only measured by the macroeconomic development level of a region, but also by the quality of interactions within a region [27,28].The existing literature, however, tends to focus on evaluating innovation based on key economic performance indicators, such as patent generation, employment rate, and commercialization of research, while neglecting the assessment of the quality of dynamic interactions among collaborating regions [29,30].

Conceivably, 3 key factors have been identified to motivate regional collaborative innovation to improve efficiency: effective allocation of innovation resources (such as talent, funds, technology, and information), the collaborative efforts

of innovation entities [31], as well as government support and an open innovation environment [32]. In its practical implementation, however, unequal resource allocation and cultural differences are two of the challenges it faces, which All of them lead to inefficient collaborative innovation or slow progress, requiring a rethink of the role of innovation policies to promote regional cooperation innovation [33].

## 2.2 The impact of regional education policies on regional collaborative innovation

Regional integration policies aim to eliminate trade barriers between different administrative regions, promote the free flow of resources and elements, and achieve reasonable industrial layout and collaborative division of labor [34]. The Growth Pole Theory, the Synergetics Theory, Innovation theory and others support the opinion that the role of regional policies in promoting regional collaborative innovation is crucial. Through the formulation and implementation of policies, governments can effectively coordinate innovation entities at different levels within a region, directly influencing the quality of interactions and outcomes of collaborative innovation [35].

The current focus of regional innovation policy remains centered on promoting scientific research excellence and the application of technology in manufacturing [36]. However, these policies often fail to effectively foster UIR collaboration, thus underutilizing the potential for regional development [12,13]. Within the framework of regional policy, education systems are a crucial component, and regional education policies are increasingly recognized as an important tool for advancing UIR cooperation [29].

Globally, regional education integration policies have proven effective in driving UIR collaborative innovation. These policies play two key roles: on one hand, they promote the development of talent and research capabilities; on the other hand, they ensure that educational resources align with industry needs, thus mobilizing both intellectual and industrial resources to provide the momentum necessary for regional economic transformation [37]

Experiences from around the world, particularly in developed countries, suggest that regional education policies have promoted collaborative innovation in the following three aspects:

First, Regional education policies enhance innovation capacities by aligning educational output with regional industry needs. Case studies by Daimer et al. [38] found that research universities and application-oriented universities in Germany closely engage with local businesses to meet regional industrial needs, thereby improving innovation capacities for both. Using interviews, OECD [13] revealed that in the innovation hubs of Ireland, and the city of Kyoto in Japan, governments guide university research to align with business needs. This precise alignment uncovers the latent innovation needs of businesses while making university research more application-oriented, thus driving rapid growth in regional patent numbers.

Second, regional education policies enhance the efficiency of technology transfer by fostering the development of industrial clusters. Clusters significantly influence innovation outcomes through agglomeration and network effects [39,40]. For instance, the "Cluster Policy Steering Group" in the UK, led by Lord Sainsbury, successfully brought together industry groups and universities, facilitating the flow of knowledge, technology, and talent. This collaboration accelerated the commercialization of scientific and technological achievements, thereby boosting regional innovation efficiency [29]. Similarly, qualitative research on the innovation network in Zhongguancun, China, demonstrates that collaboration between industry, universities, and public research institutes is a key mechanism for transferring and diffusing academic research and knowledge [41]. In Japan, Motohashi et al. [42] found that specialized industrial clusters, such as the robotics industry in Osaka, significantly fostered collaboration between industry and universities, enhancing innovation productivity.

Third, regional education policies promote the formation of innovation ecosystems with variety services. Through a previous literature review, desk research, and a limited number of supplementary telephone interviews, studies have shown that in the Nordic countries' UIR collaboration model, the establishment of the REGLAB platform which is centered around meetings, workshops, and other functions promotes resource utilization efficiency among universities, industries, and

research institutions [43]. Case studies and quantitative research by Ramos-Vielba et al. [44] show that regional policies in Andalusia, Spain, support the regional innovation ecosystem by promoting training and internships, conducting research projects, offering consulting services, and establishing hybrid R&D centers for UIR collaboration.

Despite the widespread belief that regional education policies significantly contribute to collaborative innovation, some argue that these policies have negative impacts, primarily due to challenges in their implementation. Pinto [45], through focus group interviews and qualitative assessments, found that institutional hijacking—the monopolization of the innovation policy process by a small group of dominant regional actors—led to resource monopolization and reduced innovation efficiency. Lindqvist [43] noted in a case study that overly innovative policies were initially rejected by a broad audience, and while the early effects on collaborative innovation were poor, they gradually improved as the policies were promoted. Berthold et al. [46] also observed that universities lacked "corporate citizenship," which resulted in low participation in UIR collaboration and poor outcomes. Other contributing factors include unclear responsibilities, weak stakeholder engagement, and inadequate resources [47].

The impact of regional education policies exhibits significant heterogeneity. Variations in economic development, industrial structures, and academic disciplines across regions result in differing outcomes for innovation cooperation. Felder-Stindt [48] used qualitative comparative analysis to conclude that EU regional policy instruments, such as the European Territorial Cooperation programs, have led to distinct models of Europeanization. In the Nordic countries, the acceptance of the "smart specialization" program varied, with Finland actively participating, while Sweden and Denmark expressed skepticism [43]. Daimer et al. [38] found through case studies that UIR innovation activities were more intensive in economically stronger regions and universities with more applied disciplines, such as engineering. By contrast, the innovation potential in social sciences and humanities disciplines has not been fully realized compared to the natural sciences. Similarly, Warren [49], after surveying 75 U.S. University Technology Transfer Offices (TTOs), found that regional policies often replicate successful models (e.g., Silicon Valley, Boston Route 128 Corridor) in the allocation of resources for technology transfer. These policies tend to prioritize institutions with visible innovation potential but fail to account for environmental differences among TTOs, leading to decreased innovation efficiency.

The mechanisms behind these impacts primarily involve financial incentives, platform construction, and intellectual property protection. Financial incentives are the most widespread approach. Between 2004 and 2008, the South Korean government invested $14 billion to enhance the regional impact of higher education institutions (HEIs). This funding was used to acquire advanced research equipment and attract top-tier talent, thus providing the material foundation for R&D activities. Similar initiatives include Finland's Centre of Expertise program (with a turnover of 500 million euros) and France's poles of competitiveness [29]. These programs provided financial support for market-oriented and commercialization operations, accelerating the process of taking scientific achievements from the laboratory to the market. According to OECD [13], government funding and tax incentives in Austria significantly improved regional industrial innovation capabilities. Cooperation platforms also play a pivotal role. Potts [50] analyzed how UK regional education policies, in collaboration with EU programs, fostered industry-academia cooperation through the establishment of collaborative innovation platforms. These platforms, integrating resources from universities, businesses, and governments, formed relatively stable cooperation networks. Additionally, specialized institutions, such as the 'knowledge houses' in the North East of England, provide support primarily to small and medium-sized enterprises [29]. Furthermore, Hou et al. [51] used a Difference in Difference model (DID) and found that intellectual property plays a significant mediating role in facilitating innovation cooperation.

These international cases illustrate how regional education integration policies can significantly enhance collaboration efficiency among universities, enterprises, and research institutions within a region, driving regional economies to innovate. These findings suggest that effective policy support, clear goals, and flexible coordination mechanisms are necessary for the successful implementation of such policies, which provide valuable insights into improving China's regional education integration policies.

## 2.3 Regional education integration policy and uir collaborative innovation in the Yangtze River Delta

Regional integration has become a crucial part of China's national development strategy, particularly the integration of the Yangtze River Delta (YRD) located in Eastern China, which drives China's economic development and reshapes global industrial chains [52,53]. Compared with the central and western regions in China, the impact of eastern integration is more significant, particularly in technological innovation and economic development [54]. Especially in the Yangtze River Delta in eastern region, which is home to a number of higher education institutions, research institutes, and innovative enterprises, regional education integration policies have become increasingly prominent as part of the national innovation-driven development strategy. In 2019, the YRD urban agglomeration had 413 higher education institutions and 61,000 high-tech enterprises [55]. These strong economic foundations and abundant educational resources provide strong support for regional innovation and UIR cooperation.

With the advancement of the national regional integration strategy, the education integration policy in the Yangtze River Delta has deepened. Guidelines on Further Promoting Educational Reform and Cooperative Development in the Yangtze River Delta Region (hereinafter referred to as the Guidelines) issued by the Ministry of Education in 2014 marked a milestone in the region's education integration. This policy emphasized the integration of educational resources and the enhancement of the collaborative innovation capabilities of universities, research institutions, and local enterprises to promote the transformation of scientific and technological achievements and industrial upgrading, which became the core driving force behind UIR cooperation innovation in the region, accelerating the integration process and elevating regional education integration to a new stage.

Recent findings in the literature increasingly underscore the pivotal role of educational policy in shaping regional innovation ecosystems in China. In particular, studies highlight how education policies are being leveraged not only to cultivate high-level talent and academic excellence, but also to promote industry-oriented research and cross-sectoral collaboration. These developments align with a broader global trend in which regional innovation is no longer driven solely by science and technology policies, but increasingly depends on the integration of education, economic, and industrial strategies. Within China, especially in regions like the Yangtze River Delta, emerging collaborative models show how universities and enterprises are engaging more systematically in knowledge transfer, technology commercialization, and joint R&D, all facilitated by supportive regional education integration policies. These insights provide an analytical foundation for understanding how policies are translated into practice in specific contexts such as industrial parks, university-enterprise alliances, and innovation platforms.By providing technical assistance to companies and real research and development demands to universities, the YRD, for example, has successfully promoted the transformation of university research achievements and technological innovation into enterprises [56]. Additionally, UIR integration projects in the region continue to expand, including the establishment of industrial parks in Nanjing and Hangzhou, which support innovation cooperation. The Yangtze River Delta's scientific and technological achievement transformation rate exceeded 30%, significantly higher than the national average, according to the 2020 Yangtze River Delta Science and Technology Innovation Development Report.

Although the education integration policy has played a positive role in promoting UIR collaboration, it still faces several challenges and issues. Regional innovation cooperation remains insufficient, and the policy effects vary significantly across provinces and cities [57]. For example, the comprehensive innovation efficiency of cities in the Yangtze River Delta (YRD) remains relatively low, indicating that resource allocation gaps still exist [20]. This gap is primarily due to fragmented policies or poor implementation, which impedes innovation development [35]. Therefore, the government needs to focus more on coordination and integration when implementing policies. To address these issues, policy improvements and a deeper analysis of the effects of heterogeneity are urgently needed.

## 2.4 Summary

Although existing research acknowledges the influence of regional education policies on collaborative innovation, several gaps remain. Most notably, few studies adopt quantitative approaches to directly measure the outcomes of

university-industry-research (UIR) cooperation. In addition, the dynamic interactions among collaborating regions and the mediating role of endogenous factors are often overlooked. While regional heterogeneity is sometimes mentioned, it is rarely explored in depth, especially in terms of how economic foundations, educational capacities, and policy execution vary across regions. Moreover, the majority of current research centers on developed countries, with limited examination of China's unique regional education policy dynamics under its fast-changing socio-economic conditions.

This study addresses these gaps by focusing on the Yangtze River Delta—a leading innovation hub in China—and applying a Difference-in-Differences (DID) model to quantitatively assess the impact of regional education integration policies on UIR collaborative innovation. The analysis considers patent co-application data among universities, enterprises, and research institutions, explores heterogeneous policy effects, and identifies internal mechanisms that mediate education policy outcomes. By grounding its analysis in the Chinese context, this paper contributes to a more nuanced understanding of how education policies function in emerging economies, offering empirical insights with international relevance.

## 2.5 Research hypotheses

### 2.5.1 The relationship between regional education integration policy and UIR collaborative innovation.
Theories such as Growth Pole Theory,Synergetics Theory and Innovation Theory suggest that regional education policies influence UIR collaborative innovation. Empirical research globally shows that these policies can have both positive and negative effects on collaborative innovation.

In China, the government-led education policy model plays a significant role in regional education integration and UIR collaborative innovation, with the Yangtze River Delta (YRD) region serving as a typical example. The government in the YRD actively integrates educational resources, promotes collaborative programs between universities, and achieves joint construction and sharing of curricula, faculty, and research facilities [55]. Furthermore, the government strongly promotes industry-university-research cooperation by setting up special funds and establishing demonstration bases, encouraging universities and enterprises to jointly tackle key industrial technologies to rapidly convert university research results into productive forces [56].

Liu [23] compares different regions in China and finds that the YRD has the highest regional innovation efficiency, followed by the Beijing-Tianjin-Hebei region, while the Pearl River Delta region lags behind. However, Yang et al. [20] also note that the comprehensive innovation efficiency of cities in the YRD remains relatively low. Based on these findings, we propose the following hypothesis:

H1: Regional education integration policy significantly affects university-industry-research (UIR) collaborative innovation.

### 2.5.2 The heterogeneous impact of regional education integration policy.
The Disparities in economic and educational resources across regions result in significant variations in policy effects across regions, according to the Regional Disparities Theory. A large number of cases proved that the promotion effect of regional education integration policy on UIR collaborative innovation is more significant in regions with a stronger economic base and richer educational resources, and vice versa [43,48].

In China, regional development is marked by significant imbalances, with notable differences in economic and educational resources within the Yangtze River Delta (YRD) region. For instance, cities such as Nanjing in Jiangsu Province and Hangzhou in Zhejiang Province are economically prosperous, with concentrated educational resources, numerous universities and research institutions, and advanced research facilities, attracting a large pool of high-quality talent [58]. In contrast, cities in Anhui Province, such as Bozhou, Suzhou, and Huaibei, are relatively lagging behind in both economic development and educational resources. Due to infrastructure limitations and resource shortages, these cities have weaker capabilities to attract top-tier talent and high-quality educational resources [59]. However, some studies report the opposite; Qin [60] found that regional policies in more developed cities like those in the YRD and Pearl River Delta tend to have a lower impact on enhancing UIR technological innovation efficiency.

Therefore, based on the regional imbalances in China's development and the internal diversity of the YRD, we propose the following hypotheses:

**H2a**: The impact of regional education integration policy on UIR collaborative innovation significantly differs across regions with different economic foundations and educational resource levels.

The Open Innovation Theory suggests that patents with high innovation intensity often rely on external resources and knowledge flows, especially through cooperation between regional and external innovation entities. Regional education integration policy, by providing collaboration platforms, promoting knowledge sharing, and facilitating technological transformation, plays a key role in the generation of such patents. A large literature has been established that patents are highly heterogeneous in terms of their value [61,62].

In the Yangtze River Delta, regional education policies have a significant impact on the intensity of different types of innovation output. For high-intensity innovation outcomes, particularly disruptive and technologically advanced invention patents, strong scientific research capabilities and technical platforms are usually required. As part of the national innovation system, regional education policies provide the necessary resources and support for these high-intensity innovations. By offering collaborative research projects, technology resource-sharing platforms, and innovation alliances, the YRD policies promote the development of cutting-edge technologies and the transformation of knowledge [63]. These policies not only support technological breakthroughs but also help advance industries in key technological fields. Therefore, the following hypotheses are proposed:

**H2b**: The impact of regional education integration policy on UIR collaborative innovation varies significantly depending on the intensity of innovation output

**2.5.3 The role of internal driving factors.** China's Yangtze River Delta (YRD) represents a unique case of regional education integration and UIR collaborative innovation. Its government-led policies have driven significant progress in resource sharing and technological innovation while also facing challenges due to regional disparities. The following hypotheses are proposed based on the specific characteristics of the YRD region.

According to Rodríguez-Pose et al. [64], a region is able to rely on both internal and external sources of innovation, but that the socio-economic conditions required for each region to maximize its innovation potential are necessarily internal, since the socio-economic conditions of neighboring regions do not significantly influence local economic performance.

Internal factors mediate the impact of regional education integration policy on UIR collaborative innovation. These factors not only influence the policy's effectiveness but also amplify the policy's impact on innovation collaboration through different pathways.

According to Human Capital Theory, highly skilled talent is the core driving force of innovation. The accumulation of regional human capital significantly enhances innovation capacity and accelerates the diffusion and transformation of technology through mechanisms such as higher education and skills enhancement. Wang [58], using the Yangtze River Delta as an example, empirically analyzed the key role of high-quality human resources in technological innovation, confirming that highly skilled talent is the most valuable resource in a knowledge-based economy and plays an indispensable role in driving regional innovation.

According to Policy Network Theory, government funding plays a crucial role in promoting regional innovation. Investment in research and education by the government not only enhances the collaborative efficiency of innovation entities but also facilitates technological innovation and industrialization [65]. Qin [60] emphasized the critical importance of resource allocation in Chinese universities for innovation collaboration. Empirical research indicates that investment in education has a significant positive impact on the technological innovation efficiency of China's three major regions: the Yangtze River Delta, Beijing-Tianjin-Hebei, and the Pearl River Delta.

According to Innovation Spillover Theory, economic capital plays a vital role in driving technological innovation and industrial development. For example, Foreign Direct Investment (FDI) provides financial support to enterprises, accelerates technological innovation within the region, and enhances their competitiveness in global markets [4]. Sheng et al. [66]

conducted empirical analysis showing that in China FDI has a positive impact on regional innovation, moderated by the local absorptive capacity. Chen et al. [67] also found through empirical research that FDI significantly increases the number of patent applications from Chinese enterprises, mainly by intensifying market competition, which forces companies to upgrade and innovate their technologies, rather than relying solely on knowledge spillover effects.

According to Institutional Theory, social stability provides a stable environment for innovation activities. For instance, a high employment rate helps reduce uncertainty in collaboration, increases trust and cooperation among innovation entities, and supports collaborative innovation [68]. You et al. [69] conducted an empirical analysis indicating that flexible employment, supported by excellent enterprise IT capabilities and strict labor regulations, effectively promotes innovation inputs and outputs, making a positive contribution to the innovation development of Chinese enterprises

Therefore, the following hypotheses are proposed:

**H3a:** Human capital (higher education enrollment rate) plays a significant mediating role in the promotion of UIR collaborative innovation by regional education integration policy.

**H3b:** Financial support (government investment in research and education) plays a significant mediating role in the promotion of UIR collaborative innovation by regional education integration policy.

**H3c:** Economic capital (FDI) plays a significant mediating role in the promotion of UIR collaborative innovation by regional education integration policy.

**H3d:** Social stability (employment rate) plays a significant mediating role in the promotion of UIR collaborative innovation by regional education integration policy.

Based on the above, the study forms the following research framework, as shown in Fig 1.

## 3 Research design

### 3.1 Methodology

To empirically assess the impact of regional education integration policies on UIR collaborative innovation, we employ a Difference-in-Differences (DID) model. This quasi-experimental design leverages the staggered implementation of the Yangtze River Delta (YRD) education integration policy, treating YRD provinces as the experimental group and non-YRD provinces as the control group.. The baseline regression is in the form:

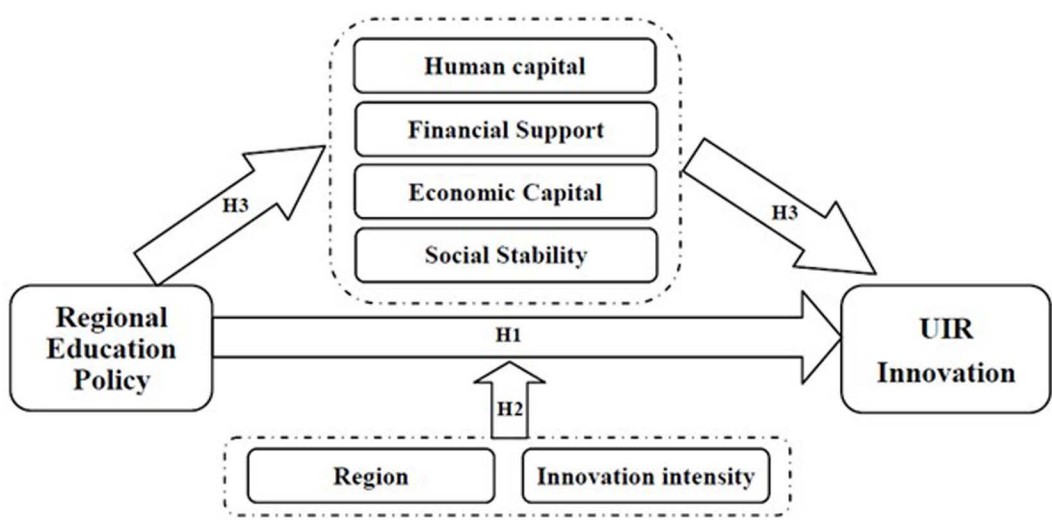

**Fig 1. Research Framework.**

$$inno_{it} = \beta_0 + \beta_1 DID_{it} + \beta_2 X_{it} + \delta_i + \gamma_t + \mu_{it}$$

Among them, *inno$_{it}$* is the innovation cooperation among industries, universities and research institutes (including the total number of inventions published, the total number of inventions authorized, the total number of utility models and the total number of designs), *DID$_{it}$* is the dummy interaction term of region and time, indicating the effect of the implementation of the Yangtze River Delta education integration policy, *X$_{it}$* are the control variables, $\delta_i$ is the individual fixed effect, $\gamma_t$ is the time fixed effect, and $\mu_{it}$ is the error term.

*DID$_{it}$* is the key regression coefficient of interest in this paper. If $\beta_1 > 0$, the education integration policy has a positive effect on UIR innovation cooperation in the Yangtze River Delta region; if $\beta_1 < 0$, the education integration policy has a negative effect on the UIR innovation cooperation in the Yangtze River Delta region; and if $\beta_1 = 0$, it would mean that the policy effect is not significant.

To account for temporal shocks—such as national economic reforms or sector-wide technological advancements—we incorporated year fixed effects (γt) into the model. This approach controls for unobserved time-specific confounders, isolating the policy effect from broader temporal trends. Additionally, we conducted event-study analyses and time-trend decomposition to disentangle dynamic policy impacts, ensuring that our results are not driven by pre-existing trends or post-treatment divergence.

Given the longitudinal nature of our panel data (2006–2021), we rigorously tested for stationarity to avoid spurious regression results. Following the reviewer's suggestion, we applied the Augmented Dickey-Fuller (ADF) test to both level and first-differenced series. We further validated the existence of long-term equilibrium relationships using the Engle-Granger two-step method, ensuring the robustness of our estimators. We further fortified our analysis through placebo tests, propensity score matching (PSM), and Regression Discontinuity Design (RDD). These methods address potential biases from omitted variables, selection effects, and spatial spillovers.

### 3.2 Data descriptions and statistics

We use panel data ranging from 2006 to 2021. Data for dependent variable*inno$_{it}$* is acquired through university-enterprise R&D alliance database in the Chinese Research Data Services (CNRDS). We obtain the provincial cooperation information of each joint patent data according to the province of each university-industry-research subject among the joint patentees; (2) For the joint patent data with three or more applicants, if there are more than two universities, take the province of the first-named university as the province of the joint patent as the record of the cooperation information of the joint patent.

For core independent variable*DID$_{it}$*, the time treatment is based on the time the Ministry of Education's Guidance is issued (1=after policy, 0=before policy), and the region treatment is similar (1=region affected by policy, 0=region not affected).

According to previous literature, control variables including the level of economic development (ln *GDP*) [29], scale of fiscal expenditure (gov) [30], degree of urbanization (urban) [31], level of industrial structure (structure) [32], urban-rural income gap (gap) [33], and level of financial development (ln *fin*) [34] were added.

Table 1 reports the main variable statistics for 466 observations.

## 4 Empirical analysis

### 4.1 Variable stationarity

Prior to conducting regression analysis, it is imperative to assess the stationarity of the variables involved. Direct regression on non-stationary original sequences may lead to spurious regression results. Therefore, this study employed the Augmented Dickey-Fuller (ADF) unit root test to determine the stationarity of each variable. The test results revealed that some of the original sequences were not stationary. Subsequently, we applied first-order differencing to these variables

**Table 1. Descriptive statistics of variables.**

| Varriable Name | Obs | Mean | S.D. | Min. | Median | Max. |
|---|---|---|---|---|---|---|
| patent | 466 | 0.432 | 0.626 | 0.001 | 0.173 | 3.762 |
| did | 466 | 0.069 | 0.253 | 0.000 | 0.000 | 1.000 |
| treat | 466 | 0.137 | 0.345 | 0.000 | 0.000 | 1.000 |
| post | 466 | 0.519 | 0.500 | 0.000 | 1.000 | 1.000 |
| gdp | 466 | 0.097 | 0.009 | 0.066 | 0.097 | 0.117 |
| urban | 466 | 0.562 | 0.138 | 0.227 | 0.550 | 0.896 |
| gov | 466 | 0.234 | 0.129 | 0.084 | 0.210 | 1.291 |
| structure | 466 | 0.474 | 0.095 | 0.298 | 0.466 | 0.839 |
| gap | 466 | 0.899 | 0.216 | 0.496 | 0.865 | 1.740 |
| fin | 466 | 0.031 | 0.012 | 0.014 | 0.028 | 0.081 |
| sci_edu | 466 | 0.041 | 0.017 | 0.014 | 0.037 | 0.159 |
| sch | 466 | 0.019 | 0.006 | 0.006 | 0.019 | 0.042 |
| fixed | 466 | 0.607 | 0.245 | 0.267 | 0.541 | 1.753 |
| emp | 466 | 0.566 | 0.064 | 0.000 | 0.562 | 0.723 |
| fdi | 466 | 2.224 | 2.015 | −1.281 | 1.847 | 12.099 |

and conducted the ADF test again. The results in Table 2 indicated that all first-order differenced sequences rejected the null hypothesis of the presence of a unit root, confirming that they were all stationary sequences of order one.

Given that not all original sequences were stationary, we further conducted a cointegration test to verify the existence of a long-term cointegration relationship among these variables. Using the Engle-Granger Two-Step Method to test the unit root of the residuals, the results in Table 3 showed that the residual sequence was stationary, thereby confirming the cointegration relationship among the variables.

## 4.2 Benchmark model regression

The estimation results of the benchmark models are presented in Table 4. Column (1) shows the estimation results without control variables, and columns (2)-(3) show the estimation results with control variables. Based on the estimation results, we can conclude that no matter how many control variables are added, the estimated coefficients of the policy variable DID remain significantly positive. As a result, education integration has a significant positive impact on

**Table 2. Augmented Dickey-Fuller (ADF) unit root test.**

| Variable | Level | ADF statistic | level | ADF statistic | result |
|---|---|---|---|---|---|
| patent | Origin | −1.5235 | First difference | 17.3713*** | stationary |
| gdp | Origin | 14.4690*** | First difference | 14.2256*** | stationary |
| urban | Origin | −0.3672 | First difference | 15.6548*** | stationary |
| gov | Origin | 12.8064*** | First difference | 14.3991*** | stationary |
| structure | Origin | 5.4705*** | First difference | 11.7034*** | stationary |
| Gap | Origin | 7.2679*** | First difference | 16.3939*** | stationary |
| Fin | Origin | 4.4091*** | First difference | 26.5543*** | stationary |

**Table 3. Engle-Granger (EG) two-step cointegration test.**

| Variable | Level | ADF statistic | result |
|---|---|---|---|
| E | origin | −4.051*** | stationary |

**Table 4. Fixed effects model results.**

| VARIABLES | (1) patent | (2) patent | (3) patent |
|---|---|---|---|
| DID | 0.539*** | 0.452*** | 0.349*** |
| | (6.26) | (5.83) | (6.12) |
| Gdp | | 0.081 | 0.364*** |
| | | (0.63) | (4.65) |
| Urban | | −0.022*** | 0.009 |
| | | (−7.29) | (1.33) |
| Gov | | −2.126** | −0.714 |
| | | (−2.17) | (−1.34) |
| Structure | | | 0.010*** |
| | | | (4.60) |
| Gap | | | 2.849*** |
| | | | (4.81) |
| Fin | | | −0.076*** |
| | | | (−2.98) |
| Constant | 0.768*** | 1.893 | −5.812*** |
| | (78.01) | (1.33) | (−4.14) |
| Observations | 466 | 466 | 466 |
| R-squared | 0.590 | 0.626 | 0.677 |
| Number of groups | 31 | 31 | 31 |
| Region fixed effect | YES | YES | YES |
| Year fixed effect | YES | YES | YES |

Standard errors in parentheses *$p < 0.1$, **$p < 0.05$, ***$p < 0.01$.

UIR innovation cooperation in Yangtze River Delta, and the estimated impact of the policy variable DID is more accurate with the addition of control variables. With control variables added, the estimated coefficient of the policy variable DID decreases, indicating that the policy's estimation effect will be more accurate.

Based on the results of control variables, economic development has a significant positive impact on UIR cooperation. This can be explained by how improvements in economic level leads to the construction of innovation service platforms and intermediary platforms, as well as the improvement of infrastructure and R&D equipment, thereby attracting and gathering innovation resources and fostering innovation collaboration. The influence of industrial structure on innovation cooperation is significantly positive, indicating that industrial agglomeration brought about by upgrading industrial structures creates not only a stable, sustainable environment for industry-university-research collaboration, but also a perfect institutional environment and a variety of application scenarios. The significant negative impact of financial development on innovation cooperation reflects, on the one hand, the aggravation of information asymmetry caused by the imperfection of the financial market, and on the other hand, capital being out of reality and turning to the virtual world, as well as idle money in the financial system, which does not flow to the real economy [29]. The scale of financial expenditure has a negative impact on innovation cooperation, suggesting that excessive government intervention leads to market forces not being fully utilized, and the allocation efficiency of innovation resources decreases, which results in a decline in innovation cooperation activities [30].

### 4.3 Parallel trend test

The parallel trend assumption is a necessary precondition for the difference-in-differences method to obtain unbiased estimation results, i.e., it requires that the experimental group and the control group have basically the same trend of change

before the policy is implemented, otherwise the effect of policy implementation may be overestimated or underestimated. In this paper, we use the event study method to test the parallel trend assumption and construct the following model:

$$\text{inno}_{it} = \beta_0 + \sum_{n=-s}^{s} \beta_1 \text{DID}_{it}^k + \beta_2 X_{it} + \delta_i + \gamma_t + \mu_{it},$$

where i and t denote province and year, respectively. $\text{DID}_{it}^k$ denotes the policy dummy variable with s denoting the specific year in which the policy was issued, and t denoting the sample year; when t-s=k(k=−7,······,7), $\text{DID}_{it}^k = 1$, otherwise $\text{DID}_{it}^k = 0$. In this paper, the first period before the implementation of the Guidance is the base group. If $\text{DID}_{it}^{k-7}$ to $\text{DID}_{it}^{-2}$ is significantly different from zero, then the model is able to satisfy the parallel trend assumption; if $\text{DID}_{it}^0$ to $\text{DID}_{it}^7$ are not significantly different from zero, it indicates that there is a policy effect of the Yangtze River Delta education integration policy.

Fig 2 demonstrates the estimated coefficients of the dummy variables and their 95% confidence intervals. Since the period before the implementation of the policy is set as the benchmark group, there are no data for the −1 period in the figure. This figure shows that overall the impact of the education integration policy on Yangtze River Delta UIR innovation cooperation is basically zero before the policy was implemented. To enhance the degree of integration and synergy of the Yangtze River Delta, relevant policies and measures were implemented before the release of the Guidelines. In Fig 1, we can see that the education integration policy already had a certain positive promotional effect 1–2 years before it was implemented. In the model, the coefficients are significantly greater than zero after the release and implementation of the Guidance, and they show an increasing fluctuating trend from period 0–7. This indicates that policy implementation has a dynamic continuity, which can promote the Yangtze River Delta regional UIR innovation cooperation for a longer period of time.

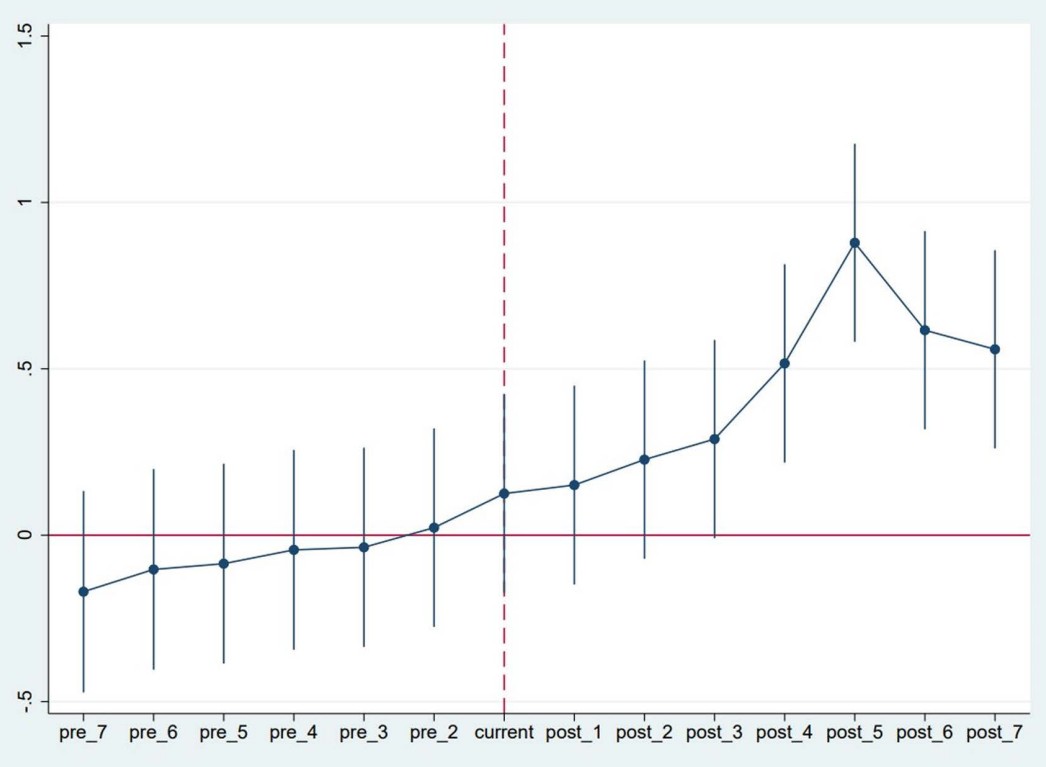

**Fig 2. Parallel Trend Analysis.**

To further validate the parallel trend assumption, we conducted a joint F-test on the pre-treatment time dummy variables (pre_7 to pre_2). The results of this test are as follows:

(1) pre_7=0, (2) pre_6=0, (3) pre_5=0, (4) pre_4=0, (5) pre_3=0, (6) pre_2=0

$F(6,430)=0.24$, Prob>$F$=0.9632

The joint F-test statistic is 0.24, with a p-value of 0.9632, which is not statistically significant. This result indicates that the trends of the experimental and control groups were not significantly different before the policy implementation, thus supporting the parallel trend assumption.

In the model, the coefficients are significantly greater than zero after the release and implementation of the Guidance, and they exhibit an increasing fluctuating trend from period 0–7. This suggests that policy implementation has a dynamic and sustained effect on promoting Yangtze River Delta regional UIR innovation cooperation.

## 5 Robustness test and heterogeneity analysis

### 5.1 Robustness tests

**5.1.1 Placebo test.** To address potential omitted variable bias and ensure the robustness of our findings, we conducted a permutation-based placebo test following the methodology proposed by Chetty et al. (2009) and Imbens & Rubin (2015). This test aims to verify whether the observed policy effect could plausibly arise from unobserved factors or confounding policies rather than the regional education integration policy itself.

We constructed 500 counterfactual experiments by randomly assigning the treatment status (i.e., designating provinces as "treated" or "untreated") while preserving the original proportion of treated units (4 out of 31 provinces). This randomization ensures that any systematic differences between the actual treated and control groups are nullified in the simulated datasets. For each simulated dataset, we re-estimated the baseline Difference-in-Differences (DID) model:

$$patent_{it} = \beta_0 + \beta_1 DID_{it} + \beta_2 X_{it} + \delta_i + \gamma_t + \in_{it}$$

where $X_{it}$ includes control variables (e.g., GDP, urbanization rate), and $\delta_i$ and $\gamma_t$ represent province and year fixed effects.

We extracted the coefficient ($\beta_1$) and its standard error from each regression to compute the t-statistic ($t=\beta_1/SE(\beta_1)$). The distribution of these t-statistics across 500 permutations was then plotted against the actual t-value observed in the baseline model.

Fig 3 displays the kernel density plot of the placebo coefficients. The distribution of the simulated t-values centers around zero and follows a near-normal distribution, with 95% of estimates falling within the range [−0.20,0.20]. In contrast, the actual policy effect ($\beta_1=0.349$) lies far in the right tail of this distribution, with a t-value of 6.12, which exceeds the 99th percentile of the placebo distribution. This stark divergence indicates that the observed effect is unlikely to be driven by chance or unobserved confounders.

By randomizing treatment assignment, this approach inherently accounts for all time-invariant province characteristics and time-varying factors shared across provinces. Crucially, it also mitigates concerns about contemporaneous policies affecting the Yangtze River Delta (YRD) provinces. If other policies systematically influenced the YRD during the study period, their effects would be captured in the placebo distribution, as these policies would remain active in the randomly assigned "treated" provinces across permutations. The absence of significant coefficients in the placebo distribution thus rules out such confounding effects.

**5.1.2 Removing the impact of COVID-19.** The sample selected for this study includes the years 2020 and beyond, and the COVID-19 pandemic occurred in 2020, which had a global impact. Public health emergencies are strictly exogenous external shocks to the macro economy, which may lead to endogeneity of variables omitted if this factor is not taken into account. After excluding the samples in 2020 and beyond, the paper conducts the same regression in Table 4, which is shown in Table 5. The core explanatory variable DID remain significant. This is consistent with the previous conclusion, indicating that the results are robust.

**5.1.3 PSM kernel density analysis.** In order to improve the quality of the sample and the credibility of the results, this paper firstly adopts PSM (propensity score matching) to match the samples, with the caliper value limited to 0.01 and

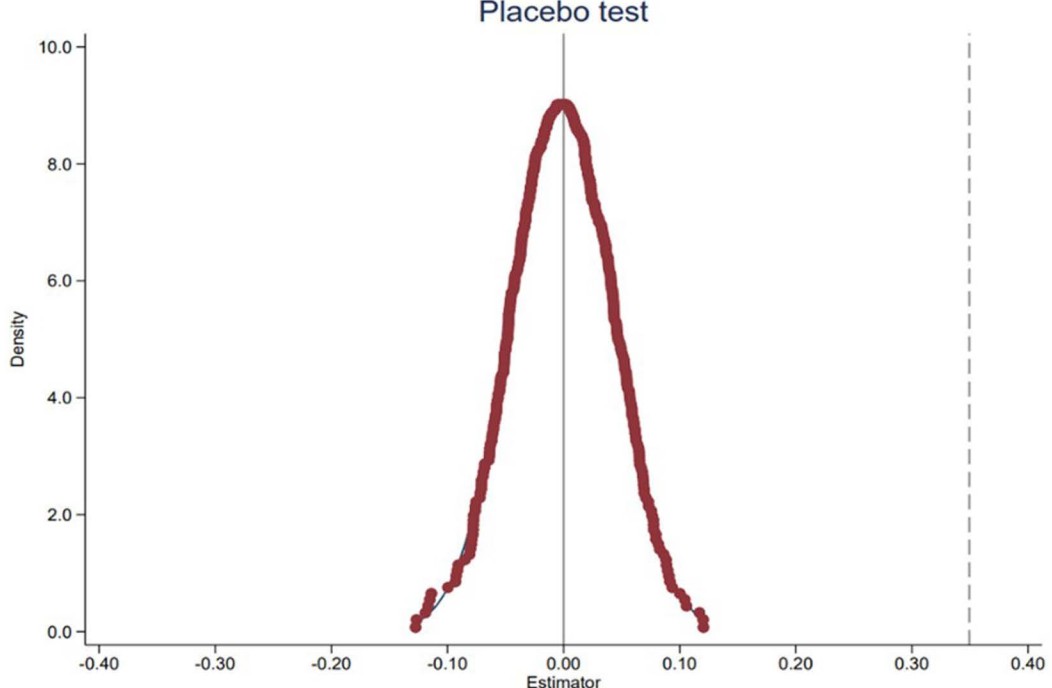

**Fig 3. Placebo Test Coefficient Kernel Density Plot.**

**Table 5. Placebo Test---Removing the impact of COVID-19.**

| VARIABLES | Excluding COVID effects patent | Excluding COVID effects Patent |
|---|---|---|
| DID | 0.464*** | 0.294*** |
| | (5.49) | (5.21) |
| gdp | | 0.421*** |
| | | (5.70) |
| urban | | −0.005** |
| | | (−2.06) |
| gov | | −2.145*** |
| | | (−5.55) |
| structure | | 0.006** |
| | | (2.44) |
| gap | | 2.262*** |
| | | (3.44) |
| fin | | −0.062 |
| | | (−1.63) |
| Constant | 0.853*** | −4.481*** |
| | (95.96) | (−4.08) |
| Observations | 404 | 404 |
| R-squared | 0.565 | 0.663 |
| Number of groups | 31 | 31 |
| area | YES | YES |
| year | YES | YES |

the mode of nearest-neighbor matching 1:1. The covariate differences between the samples before and after matching are shown in Table 6, and it can be concluded from judging the changes of the bias before and after the matching that the differences of all covariates have been significantly reduced, and the t-tests have changed from the original significant to non-significant. Significant, that is, the covariate similarity between the samples has been continuously improved, and the sample selection error has been alleviated. The PSM-matched samples are regressed again, and it can be seen that the core explanatory variable DID still has a significant positive coefficient at the 1% significance level, which is consistent with the conclusion of the benchmark regression in the previous section, and the results are robust.

**5.1.4 Regression discontinuity design analysis.** To address potential spillover effects and enhance the robustness of our findings, we expanded the Regression Discontinuity Design (RDD) analysis across multiple time periods. Building on the original framework that utilized the 2014 policy implementation as the discontinuity threshold, we conducted counterfactual tests by artificially shifting the policy year forward (2013) and backward (2015–2019). This approach allowed us to evaluate whether the observed policy effects were uniquely tied to the actual implementation year or confounded by broader temporal trends or spillovers.The RDD results, presented in Tables 7 and 8, demonstrate three key insights. First, the treatment group (Yangtze River Delta provinces) exhibited a statistically significant positive effect at the 2014 discontinuity (coefficient = 0.949, $p < 0.01$), confirming the policy's targeted impact on UIR collaborative innovation (Table 8). In contrast, the control group (non-YRD provinces) showed no significant effect (coefficient = −0.030, $p > 0.1$), ruling out spillover contamination in the immediate post-policy period.

Second, counterfactual tests for pseudo-policy years (2013 and 2015) yielded insignificant coefficients (0.151 and 0.260, respectively; Table 9), reinforcing the uniqueness of the 2014 policy anchor.

**Table 6. PSM Kernel Density Analysis.**

| VARIABLES | (1) patent | (2) patent |
|---|---|---|
| DID | 0.162** | 0.570*** |
|  | (2.21) | (4.28) |
| gdp |  | 0.631 |
|  |  | (0.91) |
| urban |  | 0.127*** |
|  |  | (5.17) |
| gov |  | 8.004** |
|  |  | (2.12) |
| structure |  | −0.038*** |
|  |  | (−4.63) |
| gap |  | 14.463*** |
|  |  | (4.95) |
| fin |  | −0.187*** |
|  |  | (−3.73) |
| Constant | 1.272*** | −23.394* |
|  | (52.91) | (−1.97) |
| Observations | 61 | 61 |
| R-squared | 0.755 | 0.929 |
| Number of groups | 18 | 18 |
| area | YES | YES |
| year | YES | YES |

**Table 7. Regression Discontinuity Design Analysis.**

| VARIABLES | The treatment group | The control group |
| --- | --- | --- |
| | Patent | patent |
| lwald | 0.949*** | −0.030 |
| | (5.57) | (−0.78) |
| Observations | 64 | 402 |

**Table 8. RDD Discontinuity Uniqueness Test.**

| VARIABLES | Pseudo-policy 2013 | Pseudo-policy 2015 |
| --- | --- | --- |
| | Patent | patent |
| lwald | 0.151 | 0.260 |
| | (0.60) | (1.13) |
| Observations | 64 | 64 |

**Table 9. Extended RDD Analysis by Policy Timing.**

| Year | Treatment Group | Control Group |
| --- | --- | --- |
| 2014 | 0.949*** | −0.03 |
| | −5.57 | (−0.78) |
| 2015 | 0.26 | −0.025 |
| | −1.13 | (−0.60) |
| 2016 | 0.282 | 0.021 |
| | −0.97 | −0.48 |
| 2017 | 0.317 | 0.064 |
| | −1.06 | −1.34 |
| 2018 | 0.355 | 0.088 |
| | −1.2 | −1.64 |
| 2019 | 0.265 | 0.082 |
| | −0.93 | −1.36 |

To address the concern about potential spillovers in later years, we extended the analysis to annual RDD comparisons from 2015 to 2019 (see S9 Table in Supporting information). The results consistently showed significant policy effects for the treatment group across all post-2014 years (e.g., 2015: 0.260; 2016: 0.282; 2017: 0.317; p < 0.1), while the control group remained unaffected (coefficients ranging from −0.025 to 0.082, all p > 0.1). This temporal granularity confirms that the policy's innovation-enhancing effects persisted without diffusing to non-target regions, even as the YRD integration deepened.

The stability of these estimates—coupled with the absence of significant discontinuities in the control group—provides robust evidence against spillover effects over time. Collectively, the expanded analysis strengthens our conclusion that the regional education integration policy had a localized, sustained impact on UIR collaboration in the YRD, with no evidence of spatial or temporal spillovers compromising the control group's comparability.

**5.1.5 Time trend effects analysis.** To exclude the potential influence of time trend effects on our findings, we adopted two approaches for verification. Firstly, we included the time trend term as a control variable in the regression model and observed whether the significance of the core variable did persisted after accounting for the time trend. The results in Table 10 demonstrated that the coefficient of did remained significant even after incorporating the time trend term,

**Table 10. Time Trend Effects Analysis.**

| VARIABLES | With Time Trend Effects | Trend Decomposition of DID |
|---|---|---|
| | Patent | patent |
| did | 0.331*** | 0.508*** |
| | (5.50) | (12.05) |
| gdp | 0.215*** | 0.236*** |
| | (12.72) | (4.04) |
| urban | 0.011 | 0.012 |
| | (1.64) | (1.67) |
| gov | −0.828 | −0.701 |
| | (−1.47) | (−1.29) |
| structure | 0.009*** | 0.008*** |
| | (3.17) | (3.20) |
| gap | 2.864*** | 2.729*** |
| | (4.56) | (5.03) |
| fin | −0.113*** | −0.085*** |
| | (−3.86) | (−3.37) |
| Constant | −173.091*** | −4.443*** |
| | (−8.82) | (−4.02) |
| Observations | 466 | 466 |
| R-squared | 0.662 | 0.696 |
| Number of groups | 31 | 31 |
| area | YES | YES |
| year | YES | YES |

indicating that its results were not influenced by the time trend. Secondly, we interacted did with multiple yearly dummy variables and included the interaction terms as control variables in the regression model. The results similarly confirmed the significance of did, further validating the robustness of our findings.

**5.1.6 Leave-one-out analysis.** To ensure our results are not driven by idiosyncratic features of a single province, we conducted a leave-one-out (LOO) analysis, iteratively excluding one of the four YRD provinces (Shanghai, Jiangsu, Zhejiang, Anhui) and re-estimating the model. Across all four permutations, the did coefficient remains statistically significant ($p < 0.01$) and stable in magnitude (0.069*** to 0.491***). The robustness of did across subsamples suggests that no single province disproportionately drives the policy effect. For example. The policy effect is not an artifact of regional outliers, reinforcing the generalizability of our findings.

## 5.2 Heterogeneity analysis

**5.2.1 Heterogeneity analysis by region.** According to the heterogeneity analysis presented in Table 11, the effects of regional education integration policies on UIR innovation cooperation differ significantly between Shanghai, Zhejiang, Jiangsu, and Anhui which compose the Yangtze River Delta region. In terms of economic development as measured by GDP in 2021, Jiangsu's GDP reached 11.63 trillion RMB, followed by Zhejiang with 7.35 trillion RMB, Shanghai with 4.35 trillion RMB, and Anhui with 4.29 trillion RMB. Based on the DID coefficients, Zhejiang and Jiangsu have significantly positive policy effects. With a coefficient of 1.199, Jiangsu has the strongest policy effect on innovation cooperation. In contrast, Shanghai has a positive but non-significant coefficient, indicating that the incremental effect is relatively low. This may be due to the fact that Shanghai's UIR cooperation was already mature before the implementation of the policy. In Anhui, however, the policy effect is negative and non-significant, indicating that the policy has not yet produced significant

**Table 11. Heterogeneity Analysis by Region.**

| | Shanghai | Zhejiang | Jiangsu | Anhui |
|---|---|---|---|---|
| VARIABLES | patent | patent | patent | patent |
| DID | 0.032 | 0.186*** | 1.199*** | −0.007 |
| | (1.68) | (3.32) | (6.24) | (−0.55) |
| gdp | 0.362*** | 0.459*** | 0.443*** | 0.377*** |
| | (8.07) | (6.83) | (6.17) | (7.39) |
| urban | −0.013* | −0.015** | −0.007 | −0.014** |
| | (−1.94) | (−2.67) | (−1.61) | (−2.47) |
| gov | −0.957 | −0.815 | −0.993* | −0.914 |
| | (−1.62) | (−1.43) | (−1.75) | (−1.63) |
| structure | 0.009*** | 0.011*** | 0.011*** | 0.009** |
| | (2.83) | (2.79) | (3.78) | (2.53) |
| gap | 1.867*** | 1.922*** | 2.285*** | 1.833*** |
| | (4.18) | (4.12) | (4.59) | (4.09) |
| fin | −0.049* | −0.035 | −0.034 | −0.049* |
| | (−1.76) | (−1.32) | (−1.21) | (−1.85) |
| Constant | −3.614*** | −4.675*** | −5.262*** | −3.713*** |
| | (−3.22) | (−3.08) | (−3.77) | (−2.95) |
| Observations | 418 | 418 | 418 | 418 |
| R-squared | 0.667 | 0.663 | 0.703 | 0.653 |
| Number of groups | 28 | 28 | 28 | 28 |
| area | YES | YES | YES | YES |
| year | YES | YES | YES | YES |

results. This may be due to the province's low economic development and relative lack of educational resources [37], which may limit the effectiveness of the policy. Indicated by the increased differences between numbers of cooperation in different regions, these findings are consistent with Huggins et al. [38] suggesting that the policy exacerbated the inequalities between inter-regional innovation cooperation, making the strong stronger and the weak weaker.

**5.2.2 Heterogeneity analysis by cooperative innovation intensity.** Patents for inventions are usually considered to have higher innovation intensity since invention patents involve new technological principles or methods, which require more R&D investment and innovation capability. Compared to invention patents, utility model patents have lower innovation intensity due to their technical requirements and innovation thresholds being lower than invention patents. A design patent usually has the lowest innovation intensity, because design mainly focuses on innovation in the visual design and aesthetics of products, and has less to do with technological breakthroughs.

Table 12 shows how regional education integration policies affect UIR collaboration under varying innovation intensities. Based on the level of protection provided by the law, invention patents can be categorized into invention publication and invention authorization. It shows that the policy promotes invention publication and invention authorization significantly, and DID coefficients are highly significant. This suggests that the policy promotes high-quality and innovation-oriented collaboration among UIR, resulting in an increase in quality and quantity of patents. In spite of the fact that utility model patent collaboration is somewhat weaker than invention patent collaboration, the policy still appears to support application-oriented innovation. Contrary to this, the policy has a significant negative impact on designs, implying that while it promotes substantial innovation cooperation, it may be less effective at promoting design-oriented cooperation. A possible explanation is that design-oriented innovation may require more market sensitivity and design flexibility to respond quickly to consumer preferences [39].

**Table 12. Heterogeneity Analysis by Cooperative Innovation Intensity.**

| VARIABLES | Invention publication Patent_pub | Invention authorization Patent_aut | utility model patent Patent_new | Design patent Patent_des |
|---|---|---|---|---|
| DID | 0.204*** | 0.084*** | 0.059*** | −0.009*** |
| | (6.19) | (6.41) | (5.50) | (−6.48) |
| gdp | 0.204*** | 0.043 | 0.146*** | 0.046*** |
| | (5.00) | (1.48) | (3.78) | (4.01) |
| urban | −0.001 | 0.005** | 0.006*** | 0.001*** |
| | (−0.42) | (2.30) | (3.11) | (3.65) |
| gov | −0.172 | −0.312*** | −0.146 | 0.068** |
| | (−0.58) | (−3.74) | (−0.92) | (2.15) |
| structure | 0.003* | 0.003*** | 0.005*** | −0.001** |
| | (1.88) | (3.69) | (5.53) | (−2.21) |
| gap | 1.437*** | 0.744*** | 0.711*** | 0.095*** |
| | (4.91) | (7.63) | (3.72) | (5.62) |
| fin | −0.049*** | −0.012 | −0.001 | 0.006* |
| | (−3.30) | (−1.69) | (−0.18) | (1.88) |
| Constant | −2.616*** | −1.198*** | −2.392*** | −0.599*** |
| | (−4.78) | (−6.34) | (−3.40) | (−5.40) |
| Observations | 464 | 457 | 445 | 235 |
| R-squared | 0.661 | 0.616 | 0.720 | 0.217 |
| Number of groups | 31 | 31 | 31 | 31 |
| area | YES | YES | YES | YES |
| year | YES | YES | YES | YES |

## 6 Mechanism analysis

To avoid the endogeneity problem from the traditional three-step method, this paper employs Jiang's [34] two-step mediation effect to examine how the independent variable affects the dependent variable. The first step is the regression of the core explanatory variables on the dependent variable, i.e., the previous benchmark regression, which is significant. In the second step, the dependent variable is replaced by the mediator variable. If the core explanatory variable still has a significantly positive effect on the mediator variable, the mediation effect exists.

Table 13 shows that the proportion of students in higher education (sch), foreign direct investment (fdi), government expenditure on science education (sci_edu), and employment rate (emp) are mediating variables that significantly contribute to the UIR innovation cooperation.

A region's higher education enrolment rate (sch) reflects its human capital. Being one of the core drivers of innovation, a highly educated labor force can not only driving technological innovation, but can also facilitate the diffusion and application of knowledge. In Table 11, it shows that the higher education enrollment rate mediates the relationship between policy and innovation cooperation, with a coefficient of 0.370, indicating a significant mediating effect consistent with previous findings [41].

As a key indicator of human capital reserves, higher education enrollment rates significantly influence the depth and breadth of university-industry cooperation (UIR). A high enrollment rate expands the talent pool in higher education, creating favorable conditions for enterprises to identify and absorb innovative talent [70]. In recent years, China's higher education enrollment rate has steadily increased, reaching 58.42% in 2020 [71]. This rising enrollment rate has enhanced universities' attractiveness in UIR collaborations, encouraging enterprises to actively seek

**Table 13. Mechanism Analysis.**

| VARIABLES | (1) sch | (2) sci_edu | (3) fdi | (4) emp |
|---|---|---|---|---|
| did | 0.370*** | 0.191* | 0.075* | 0.023*** |
|  | (6.28) | (1.95) | (1.73) | (3.50) |
| gdp | −0.003*** | 0.009*** | 3.435*** | 0.018 |
|  | (−3.00) | (15.00) | (4.70) | (1.24) |
| urban | 0.000*** | 0.000*** | 0.055 | −0.003*** |
|  | (6.97) | (3.56) | (1.58) | (−9.06) |
| gov | 0.003 | 0.094*** | 13.398*** | −0.152* |
|  | (1.52) | (37.26) | (5.97) | (−1.98) |
| structure | 0.000 | 0.000 | −0.022*** | 0.002*** |
|  | (0.99) | (0.12) | (−3.64) | (9.03) |
| gap | −0.018*** | 0.010* | −0.597 | −0.061*** |
|  | (−7.44) | (1.89) | (−0.44) | (−4.33) |
| fin | −0.001 | 0.004*** | −0.368*** | −0.000 |
|  | (−1.07) | (14.49) | (−4.90) | (−0.10) |
| Constant | 0.050*** | −0.116*** | −37.426*** | 0.529*** |
|  | (5.36) | (−12.71) | (−3.34) | (3.65) |
| Observations | 466 | 466 | 466 | 466 |
| R-squared | 0.875 | 0.856 | 0.329 | 0.358 |
| Number of groups | 31 | 31 | 31 | 31 |
| area | YES | YES | YES | YES |
| year | YES | YES | YES | YES |

cutting-edge technologies and high-potential talent, facilitating knowledge transfer and technological advancement, and driving deeper integration of academia, industry, and research [72]. Therefore, the continuous increase in China's enrollment rate provides a solid foundation for the sustainable development of UIR cooperation. Variable government expenditure on science education (sci_edu) reflects the level of government support and intervention in science education. Increased government investment improves research infrastructure and the academic environment, further facilitating effective collaboration between universities and industry. This is consistent with the recent findings in which public expenditure is an important driver in supporting science and technology innovation among UIR [42].

Financial support, especially government investment, plays a crucial role in university-industry cooperation (UIR) and has a significant positive impact. Through fiscal policies and research funding, governments help enterprises alleviate the financial burden of fundamental research, thereby reducing innovation risks and increasing their willingness to collaborate with universities. This facilitates the sharing of technology and innovation outcomes [73,74]. Additionally, government funding supports the development of research infrastructure in universities, enhancing their research capabilities and creating conditions for higher-quality collaboration, which further strengthens the depth of UIR partnerships [75,76]. In China, government funding plays a unique institutional role in driving UIR cooperation. Unlike market-driven models in many other countries, the Chinese government takes a leading role in university-industry partnerships. Through policy formulation, the establishment of special funds, and direct resource allocation, the government not only mitigates enterprise innovation risks but also actively fosters productive R&D collaboration between universities and industries [77]. Empirical research by Song [78] confirms that the Chinese government leverages funding support as a tool to promote sustainable innovation within university-industry cooperation.

Foreign direct investment (fdi) is an expression of economic capital. Results indicate that FDI plays a major role in regional innovation cooperation. This is consistent with previous research findings that increased FDI helps regional education integration policies to effectively attract more external financial support, facilitates the development and implementation of innovation projects, and benefits to improve the quality of regional innovation cooperation [43].

Foreign direct investment (FDI) significantly promotes university-industry cooperation (UIR) by enhancing corporate innovation and strengthening research and development (R&D) foundations. The introduction of advanced technology, managerial expertise, international standards, and capital through FDI not only improves firms' innovation capabilities and R&D infrastructure but also stimulates their demand to expand collaboration and deepen cooperation, thereby broadening and intensifying UIR partnerships [79,80]. In China, the role of FDI in fostering UIR cooperation has been particularly pronounced. Since the launch of the Reform and Opening-up policy in 1978, FDI inflows have surged from virtually zero to $189 billion in 2022 [81]. Foreign enterprises have not only driven growth in exports and industrial output but have also optimized resource allocation to improve innovation efficiency. Additionally, by advancing market-oriented reforms, FDI has enhanced firms' flexibility and openness to external collaboration, further strengthening their capacity and willingness to engage with universities. This process has provided a solid economic foundation for the sustainable development of UIR cooperation [82].The employment rate (emp) is a key indicator of the socio-economic stability of a region. The results show that high employment rate implies economic health and stability of the society, which becomes a driver of innovation cooperation. The same result was found in previous studies that a stable employment environment reduces social risks and enhances the sustainability of innovation cooperation [44].

Social stability is a key factor in promoting university-industry cooperation (UIR), with employment rate serving as a core indicator. High employment reduces policy uncertainty, boosts market confidence, and encourages enterprises to invest in long-term R&D and engage in deeper collaboration with universities [83]. Guimón [84] highlights that stable employment helps enterprises attract and retain R&D talent, enhancing innovation capacity and strengthening UIR cooperation. Conversely, low employment rates can lead to social instability, weakening the foundation for university-industry partnerships. Kleiner-Schaefer [85] argues that social instability hinders universities from transforming into "entrepreneurial universities," thereby obstructing knowledge transfer. Lee [83] also notes that rising unemployment forces enterprises to prioritize short-term survival, reducing investment in high-risk R&D and suppressing UIR cooperation. China's consistently stable employment landscape has provided strong support for UIR cooperation. According to the National Bureau of Statistics, China's urban registered unemployment rate remained below 5% from 2011 to 2021. Data from the Ministry of Human Resources and Social Security of China show that from 2013 to 2022, urban employment in China increased by 130 million, averaging 13 million new jobs per year. This stable employment environment has fostered UIR cooperation by facilitating better alignment between university research and enterprise innovation needs, thereby driving industrial upgrading and economic growth.

In order of the intensity of driving force, human capital is the largest, followed by government support, economic capital, and social stability. Among them, human capital quality plays a key role, and social stability is crucial for innovation cooperation. Overall, these findings emphasize the multifaceted nature of institutional innovation, present a multidimensional driver framework, and emphasize the importance of driver factors for promoting UIR collaboration.

To further validate our findings and address potential endogeneity issues, we conducted an endogenous GMM analysis. The results in Table 14, presented in the following table, confirm the robustness of our previous findings. The SYS-GMM and DIF-GMM methods were employed to ensure the reliability of our estimates. The AR serial correlation tests indicate no second-order or higher-order autocorrelation in the disturbance terms, satisfying the prerequisites for GMM regression. Additionally, the Hansen over-identification tests support the validity of all instrumental variables, further validating our model's applicability.

The significant positive correlation between the first-order lag of the dependent variable and its current value, as observed in the GMM results, highlights the "time persistence" characteristic of the dependent variable. The GMM methods effectively address this issue, providing more accurate estimates of the relationships among variables.

**Table 14. GMM Analysis.**

|  | SYS-GMM | DIF-GMM |
|---|---|---|
| VARIABLES | Patent | patent |
| L.patent | 0.9496*** | 0.8905*** |
|  | (166.29) | (82.63) |
| did | 0.0466*** | 0.0197** |
|  | (6.17) | (2.45) |
| gdp | 0.0522*** | 0.0773*** |
|  | (12.24) | (5.77) |
| urban | 0.0003 | 0.0058*** |
|  | (0.61) | (7.15) |
| gov | −0.0203 | −0.2324*** |
|  | (−0.54) | (−3.72) |
| structure | 0.0026*** | 0.0055*** |
|  | (5.72) | (5.12) |
| gap | 0.0230 | 0.4245*** |
|  | (0.84) | (6.30) |
| fin | 0.0024 | −0.0079 |
|  | (0.45) | (−1.27) |
| Constant | −0.6055*** | −1.5444*** |
|  | (−7.64) | (−6.63) |
| Observations | 431 | 431 |
| Number of id | 31 | 31 |
| AR1 | 0.002 | 0.001 |
| AR2 | 0.828 | 0.785 |
| Hansen | 0.524 | 0.681 |

## 7 Discussion

The empirical results shown in Table 4, along with the robustness tests and parallel trend test results, support Hypothesis 1. This is consistent with previous literature [13,38,43]. Innovation Systems Theory emphasizes the importance of interactions and collaboration among innovation actors in enhancing overall innovation capabilities. This study finds that regional education policies, by promoting collaboration between universities and enterprises, have increased the number of joint patent applications, reflecting the policy's key role in enhancing interactions within the innovation system. This supports the view within Innovation Systems Theory that policy serves as a coordinator and facilitator of the system, further proving the important role of regional education integration policies in optimizing innovation system structures and enhancing the overall innovation capacity of the system. Specifically, the direct support for R&D funding and the encouragement of technology transfer in the Yangtze River Delta have played a key role in facilitating joint patent applications, showing the policy's tangible impact on innovation outputs.

The empirical results in Table 10 reveal the heterogeneity of regional economic levels in affecting the impact of regional education integration policies on UIR collaboration, supporting Hypothesis 2a. Indicated by the increased differences between numbers of cooperation in different regions, these findings are consistent with Huggins et al. [86] and Felder-Stindt [48], suggesting that the policy exacerbated the inequalities between inter-regional innovation cooperation, making the strong stronger and the weak weaker. This finding can be further explained by the Growth Pole Theory, which suggests that regions with higher economic levels inherently possess more developed innovation systems and resource

allocation capabilities, making the policy more effective in these regions. In contrast, regions with lower economic levels may lack sufficient infrastructure and resources, limiting the effectiveness of the policy. For example, in more economically advanced cities like Jiangsu province, the concentration of technological expertise and business networks has amplified the benefits of regional education integration policies, leading to a higher volume of collaborative innovation. Conversely, in less developed region, the policy has had a more modest impact due to limited access to resources and innovation infrastructure. This result confirms that regional education policies amplify existing structural advantages, thus reinforcing regional disparities in innovation collaboration.

The empirical results in Table 15 highlight the heterogeneity of innovation output intensity in influencing the impact of regional education integration policies on industry-university-research (UIR) collaboration, supporting Hypothesis 2b. While prior literature has acknowledged the heterogeneity in innovation output value, empirical tests of this phenomenon are scarce. Specifically, higher-intensity innovation outputs are more strongly affected by the policy, while lower intensity outputs show weaker effects. This suggests that the policy plays a key role in driving technological innovation and knowledge transfer, particularly in promoting high-tech outputs. A possible explanation is that design-oriented innovation, which requires greater market sensitivity and flexibility to respond to consumer preferences, may not rely heavily on policy support. The policy appears to be more effective in sectors like advanced manufacturing and high-tech industries, where the push for knowledge transfer and research commercialization is more pronounced. This effect is particularly observable in technology hubs within the region, where universities and research institutions have closer ties to cutting-edge industries.

The empirical results in Table 11 reveal the internal mechanisms by which regional education integration policies impact UIR collaboration, supporting all hypotheses in Hypothesis 3. The study shows that human capital, government support,

**Table 15. Results of Leave-One-Out Analysis.**

| VARIABLES | (1) | (2) | (3) | (4) |
|---|---|---|---|---|
| | patent | patent | patent | patent |
| did | 0.426*** | 0.491*** | 0.069*** | 0.431*** |
| | (5.91) | (5.94) | (4.55) | (6.82) |
| gdp | 0.438*** | 0.399*** | 0.402*** | 0.263*** |
| | (5.39) | (5.15) | (7.01) | (3.66) |
| urban | 0.000 | 0.012 | −0.012 | 0.011 |
| | (0.03) | (1.58) | (−1.66) | (1.59) |
| gov | −0.845 | −0.772 | −0.760 | −0.746 |
| | (−1.61) | (−1.48) | (−1.26) | (−1.45) |
| structure | 0.012*** | 0.010*** | 0.010*** | 0.009*** |
| | (5.17) | (4.32) | (3.06) | (3.59) |
| gap | 2.709*** | 2.757*** | 1.979*** | 2.850*** |
| | (4.92) | (4.74) | (4.04) | (4.89) |
| fin | −0.050* | −0.065** | −0.048* | −0.103*** |
| | (−1.82) | (−2.59) | (−1.84) | (−4.16) |
| Constant | −6.061*** | −6.222*** | −4.220*** | −4.754*** |
| | (−4.25) | (−4.27) | (−2.98) | (−3.76) |
| Observations | 450 | 450 | 450 | 450 |
| R-squared | 0.667 | 0.684 | 0.689 | 0.669 |
| Number of groups | 30 | 30 | 30 | 30 |
| area | YES | YES | YES | YES |
| year | YES | YES | YES | YES |

economic capital, and social stability are all driving forces between policies and innovation cooperation. Among these, human capital is the most important driver of innovation cooperation, while social stability is the weakest driving force for innovation cooperation. These findings clarify how internal capacities—particularly human capital and government support—mediate the policy's impact on collaborative innovation outcomes.

Human capital, as the most important driving force, reflects the role of high-quality talent in promoting UIR collaboration in the region. This is consistent with prior findings that the development of higher education increases the supply of human resources, facilitates high-level innovation activities, and enhances overall innovation capacity [81,87]. This result aligns with Innovation Systems Theory, which emphasizes the central role of human capital in innovation systems. Additionally, Synergetics Theory points out that regions with high-quality human capital can more effectively coordinate various resources, promoting cooperation efficiency. In contrast, regions with less access to high-quality talent, such as those outside major metropolitan areas, often struggle to attract and retain skilled workers, which limits the effectiveness of education integration policies.

Government support, as the second most important driving force, reflects the key coordinating and incentivizing role of public funding in UIR collaboration. This is consistent with recent findings in which public expenditure is an important driver in supporting science and technology innovation among UIR [65]. This finding supports the role of government as a coordinator and facilitator in Innovation Systems Theory and echoes Growth Pole Theory, which emphasizes the role of the government in promoting regional agglomeration and innovation activities. Government funding has been particularly impactful in promoting early-stage research and development in regions, where public-private partnerships are driving innovation in emerging industries [88].

Economic capital, as the third driving force, indicates that sufficient economic resources in a region can provide the necessary support and guarantees for UIR collaboration. This is consistent with previous research findings that increased foreign direct investment (FDI) helps regional education integration policies effectively attract more external financial support, facilitates the development and implementation of innovation projects, and improves the quality of regional innovation cooperation [89]. Empirical research by Yang et al. [20] also shows that economic capital has a significant positive impact on innovation efficiency in the Yangtze River Delta (YRD) region. This result aligns with the view of resource optimization and allocation in Synergetics Theory, indicating that sufficient economic capital helps enhance cooperation efficiency and the quality of innovation outcomes. In the YRD region, cities like Suzhou, which attract significant foreign investment, benefit from a steady inflow of economic capital that supports innovation activities, while smaller cities without such investments face more challenges in sustaining innovation collaboration.

Social stability also acts as a driving force in promoting cooperative innovation through policy. The same result was found in previous studies that a stable employment environment reduces social risks and enhances the sustainability of innovation cooperation [90]. However, compared to other driving forces, social stability is the weakest driver, indicating that although social stability has some impact on UIR cooperation, its direct influence is smaller. This may be because, in regions with high stability, other factors, such as human capital and government support, are already sufficient to drive collaborative innovation. In contrast, in regions with low stability, despite potential challenges in the social environment, high human capital and government support can partially offset these negative impacts. Therefore, while social stability is important, its influence is less significant compared to other drivers within the framework of this study. While social stability has been important in maintaining long-term collaboration, its role in spurring immediate innovation activities appears to be secondary to other factors like human capital and government support.

Overall, these findings emphasize the multifaceted nature of institutional innovation, present a multidimensional driver framework, and highlight the importance of driver factors in promoting industry-university-research collaboration. The study's findings underscore the importance of tailoring regional education integration policies to local contexts, particularly in terms of economic conditions, human capital availability, and government support, to maximize their impact on innovation cooperation.

While this study focuses on the Yangtze River Delta region in China, the findings and insights derived from the empirical analysis are not limited to this region alone. The regional education integration policies analyzed in this study—designed to promote collaboration between universities, research institutions, and industries—are common to many innovation ecosystems globally. Research from countries such as Germany, Ireland, and Japan has similarly shown that policies aimed at strengthening university-industry collaboration drive regional innovation [13,38]. This mechanism is also effective in the Yangtze River Delta region, confirming its applicability across different economic contexts. The specific impact of these policies, however, varies depending on local socio-economic conditions, suggesting that other emerging economies could benefit from customizing these policies to their own needs.

Furthermore, the study's findings on the impact of regional economic levels and innovation intensity on policy effectiveness align with global research [43,48]. In regions with higher economic or innovation intensity, such as Silicon Valley (USA), the effects of regional education integration policies are more pronounced. This suggests that similar policies could be effective in other regions, especially those with varying economic conditions. For example, the technology transfer experiences in Silicon Valley further support this notion, highlighting the global relevance of the policy's impact. This suggests that China's regional education integration policies may serve as a valuable model for other developing economies seeking to enhance their innovation systems.

Additionally, the key drivers identified in this study, such as human capital, government support, economic capital, and social stability, are essential components of fostering innovation and have been widely recognized in global research [65,91]. These factors play a critical role in shaping regional innovation policies, providing valuable insights for policymakers in China and around the world. The study's findings, therefore, offer practical guidance for regions seeking to optimize their innovation systems and promote sustainable economic development.

## 8 Conclusions and implications

This study contributes to the understanding of how regional education integration policies influence UIR innovation cooperation. By using panel data from China's provinces between 2006 and 2021 and applying the DID method, we demonstrate that these policies significantly promote UIR collaboration, especially in regions with strong economic foundations and well-developed educational resources. The study contributes to the literature by highlighting the heterogeneity of policy effects across different provinces and innovation output intensities, which has been underexplored in previous research. The findings suggest that policies are more effective in regions with robust economic fundamentals and where scientific and technological innovation plays a significant role, rather than in regions driven mainly by market forces. Moreover, our mechanism analysis uncovers key mediating factors—human capital, government expenditure, economic capital, and social stability—that facilitate the positive impact of these policies. This study offers valuable insights for policymakers, emphasizing the need to design region-specific policies that can better integrate educational and industrial resources, particularly in economically diverse regions.

Based on the above research results, some policy implications are as follows:

The findings highlight the necessity for governments to integrate regional education reforms into broader innovation-driven development strategies. Particularly in high-growth regions and national tech hubs such as the Yangtze River Delta and the Guangdong-Hong Kong-Macao Greater Bay Area, educational integration policies should explicitly align with sectoral innovation priorities, embedding universities more deeply into regional industrial value chains. By aligning educational policies with long-term economic goals, governments can ensure that the talent cultivated by universities better meets the actual needs of businesses and industries, thereby improving the overall quality of human capital. Establishing collaborative mechanisms, such as university-industry research platforms and joint academic-industrial programs, can serve as effective tools to bridge the gap between education and industry. These platforms not only facilitate resource sharing and joint R&D but also provide tailored support for industrial transformation and technological upgrades. Such efforts will enable regions to better adapt to global economic changes, enhancing their international competitiveness while fostering sustainable innovation-driven growth.

Creating a supportive environment for industry-university-research cooperation is equally critical. In addition to foundational investment, governments should actively design multi-level funding incentive schemes to stimulate industry-academia cooperation. For example, offering tax breaks to enterprises engaging in long-term collaborations with universities or establishing performance-based funding linked to collaborative outputs such as patents or joint publications. Local governments should prioritize investments in improving university teaching quality, developing specialized programs, and strengthening faculty training to ensure a continuous supply of high-quality talent. To incentivize collaboration, the establishment of dedicated research and innovation funds, especially in high-priority sectors like AI and renewable energy, can play a pivotal role. Additionally, building shared technology platforms and regional innovation hubs can enhance the integration of educational and industrial resources, driving technological advancement and strengthening industrial chains. Governments should also encourage social capital to flow into research-focused universities and technology enterprises, optimizing resource allocation and accelerating the development of collaborative innovation ecosystems.

Finally, differentiated policy approaches are essential to address regional disparities. In economically weaker regions, priority should be given to improving educational infrastructure, such as building research facilities and enhancing basic research capabilities, to provide a foundation for innovation. In contrast, regions with stronger innovation ecosystems should focus on advancing high-end research platforms, fostering international scientific cooperation, and establishing cross-regional innovation alliances to consolidate their competitive edge. Policies should also prioritize high-risk, high-reward frontier research, particularly in disruptive technologies, to ensure long-term leadership in global innovation. These tailored strategies will maximize the benefits of regional education integration policies, fostering balanced and sustainable development across diverse economic contexts. Looking ahead, policymakers should also promote the digital transformation of education systems to better support innovation in strategic emerging sectors. Emphasis should be placed on developing interdisciplinary programs related to artificial intelligence, green technology, and digital manufacturing, which are critical to the evolving needs of China's innovation economy.

## 9 Limitations and further research proposals

This study has three main limitations. First, the data and policy context are primarily focused on specific regions in China, which may limit the generalizability of the findings to other regions with different economic and educational contexts. While the Yangtze River Delta region serves as a representative case, future research should expand the geographic scope to include other regions in China and compare the effects of similar regional education policies in countries with differing industrial structures and policy environments. This would help to validate the broader applicability of the results.

Second, the study focuses mainly on the short- and medium-term effects of regional education integration policies on UIR cooperation, without considering the long-term impact, particularly in light of the digital transformation of education. The rapid development of digital education policies and their interplay with university research, enterprise R&D, and social innovation could have significant long-term effects that are currently underexplored. Future studies should explore how digital education policies and technological advancements influence the sustainability and evolution of UIR collaborations in the long run.

Third, while our difference-in-differences (DID) framework rigorously tests pre-treatment parallel trends, post-treatment divergence risks persist due to three contextual factors: (1) spatially correlated economic shocks (e.g., trade wars, RMB depreciation crises) disproportionately affecting the Yangtze River Delta region; (2) policy spillover contamination from non-YRD provinces adopting similar innovation cluster policies, attenuating treatment effect estimates; and (3) evolving heterogeneity in extended post-treatment periods, exacerbated by China's shifting macro-strategies like the "dual circulation" policy. Although robustness checks using staggered DID estimators [92]-confirm medium-term effect stability, long-term interpretations beyond five years remain cautiously framed. These contextual factors, including regional economic shocks and macro-strategic shifts, warrant further research to better understand their influence on the long-term impacts of regional educational policies. Furthermore, additional investigations could explore how these factors interact

with regional disparities in innovation capacity, particularly in lower-income or less-developed areas, to better capture the nuances of policy effectiveness.Future research should explore several important avenues to deepen our understanding of the effects of regional educational policies on UIR collaborations. First, expanding the scope of the study to include other Asian economies or even a global context would provide valuable insights into the universality and boundary conditions of the findings. Comparing the effects of regional education integration policies in countries with diverse economic structures—such as India, South Korea, and Japan—could reveal whether similar mechanisms and impacts hold across different institutional and cultural contexts. Furthermore, incorporating the experiences of developed economies with advanced educational frameworks could highlight additional insights for optimizing regional education integration policies globally.

Second, investigating the long-term impact of digital education policies is essential, especially considering the rapid technological transformations shaping education worldwide. As digital teaching methods, online platforms, and AI-driven tools become more integral to education, future research should assess how these innovations affect UIR collaboration, particularly with regard to technology diffusion and industrial upgrading. Exploring how digital education policies interact with enterprise R&D and university research in the long term could offer a more comprehensive view of the sustainability of UIR collaborations in a digital economy.

Third, employing high-frequency data and heterogeneity-robust estimators—such as synthetic DID or dynamic spatial econometric models—could further refine the analysis, enabling scholars to better isolate transient shocks from sustained policy effects. These methodologies could address the biases caused by spatially correlated economic shocks or policy spillover contamination, enhancing the robustness of future studies.

Finally, exploring inter-regional policy spillovers and the impact of broader macroeconomic strategies—like China's "dual circulation" policy—could reveal new insights into the diffusion of innovation across regions and the factors that influence policy effectiveness over time. These studies could also explore how varying innovation capacities across regions, especially in lower-income or less-developed areas, impact the effectiveness of educational integration policies.

## Supporting information

**S1 Data. Dataset for empirical analysis.**
(ZIP)

**S2 Code. Stata analysis codes.**
(ZIP)

## Acknowledgments

Acknowledge Chenyu Liu from Lancaster University for insightful reviews to improve this article. The author(s) disclosed receipt of the following financial support for the research, authorship, and/or publication of this article: This research was funded by the National Education Sciences Planning of China under Grants of Research on the Integration of Higher Education in Yangtze River Delta from the Perspective of Function Driver (DIA200347).

## Author contributions

**Conceptualization:** Qi Chen, Jiawen Zhou.

**Data curation:** Qi Chen, Jiawen Zhou.

**Formal analysis:** Qi Chen, Jiawen Zhou.

**Funding acquisition:** Qi Chen.

**Investigation:** Qi Chen, Jiawen Zhou.

**Methodology:** Qi Chen, Jiawen Zhou.

**Project administration:** Qi Chen, Jiawen Zhou.

**Resources:** Qi Chen, Jiawen Zhou.

**Software:** Qi Chen, Jiawen Zhou.

**Supervision:** Qi Chen, Jiawen Zhou.

**Validation:** Qi Chen, Jiawen Zhou.

**Visualization:** Qi Chen, Jiawen Zhou.

**Writing – original draft:** Qi Chen, Jiawen Zhou.

**Writing – review & editing:** Qi Chen, Jiawen Zhou.

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
