## [Decision Letter · Decision Letter 0]

PONE-D-24-44621Can regional education integration policy promoted university- industry-research collaborative innovation? –Evidence from ChinaPLOS ONE?

Dear Dr. Zhou,

**Please note that a**
**lthough some reviewers may ask you to cite certain papers, DO NOT cite them if they are not relevant to your manuscript.**

We look forward to receiving your revised manuscript.

Kind regards,

Xu Xin

Academic Editor

PLOS ONE

Journal Requirements:

"Chen Qi

DIA200347

the National Education Sciences Planning of China

https://onsgep.moe.edu.cn/

YES"

Reviewers' comments:

Reviewer's Responses to Questions

**Comments to the Author**

1. Is the manuscript technically sound, and do the data support the conclusions?

Reviewer #1: Partly

Reviewer #2: Partly

Reviewer #3: Yes

2. Has the statistical analysis been performed appropriately and rigorously?

Reviewer #1: No

Reviewer #2: Yes

Reviewer #3: Yes

3. Have the authors made all data underlying the findings in their manuscript fully available?

Reviewer #1: No

Reviewer #2: Yes

Reviewer #3: Yes

4. Is the manuscript presented in an intelligible fashion and written in standard English?

Reviewer #1: Yes

Reviewer #2: Yes

Reviewer #3: Yes

**Reviewer #1:**

The paper investigates the impact of regional education integration policies on collaborative innovation between universities, industries, and research institutions (UIR) in China’s Yangtze River Delta (YRD). The policy was implemented in 2014 on 4 province-level administrations -- Shanghai, Jiangsu, Zhejiang, and Anhui. It uses a difference-in-difference model to assess the effectiveness of these policies. The paper finds that regional education integration policy significantly enhances UIR innovation cooperation, particularly in Zhejiang and Jiangsu, which have stronger economies and richer educational resources. The policy is especially effective in promoting high-intensity innovation, reflected in the increase of invention patents. Four internal factors mediate the success of these policies: human capital (e.g., higher education enrollment), economic capital (e.g., foreign direct investment), government support (e.g., science education funding), and social stability (e.g., employment rate).

Though the findings of the paper is intuitive and reasonable, the methodologies and evidence are questionable, which may not be able to sufficiently support its conclusion. The authors use two-way-fixed-effect difference-in-difference for causal inference. The validity of difference in difference relies on two important assumptions.

The first one is there is no spillover effect, i.e. the control group cannot be impacted by the treatment. However, since treated region is one of the largest economic hubs China. It is unbelievable that the policy will only influence universities and companies in the region but not those in other provinces. The authors also mentioned that there are some collaborations including multiple universities from different provinces, which can be an indication of the potential of spillover effect of treatment.

The second assumption is the parallel trend assumption. It requires the trend of the treatment and control groups to be parallel over time if there was the treatment. The authors test for the parallel of the pre-treatment trends. Passing the test can only prove half of the parallel trend assumption is correct. Authors should discuss how likely the post-treatment trends would still be parallel without the treatment. Given how important and special YRD region is for China, considering its cumulative advantage in science, technology, education, and governments’ discretion in economic management, it is hard to believe the trend of the region would be the same as the control group without treatment.

What’s more, the claim that the pre-treatment trends being parallel is also not convincing. First, Figure 1 clearly shows the point estimate of the difference between treatment and control is trending up before the treatment. The trend even look consistent to the trend after the treatment. The plot even makes me feel that the after treatment difference is more likely due to the preexisting trend rather than the treatment. Second, even though none of the point estimate in the pre-treatment test is significant at 95% level, the author fails to offer the joint F statistic of the 7 point estimates. When testing for parallel trend, joint F-statistics is more reliable than checking each point estimate. The authors may have realized the flaw in the test. They use PSM to create kernel density weighted average of control groups. However, they failed to show the test for pre-treatment parallel trend for the PSM-DID model. They also attempt to use instrumental variable. But the instruments are problematic, too. First order lag has been widely criticized in academia since it probably violates exclusion restrictions. i.e. instrumental variables can only influence the dependent variable through the corresponding endogenous variables.

I also observed some mistakes in the description of concepts and methods. For instance, between Row 282 and 285, “if DIDk−7 to DID−2 is significantly different from it zero, then the model is able to satisfy the parallel trend assumption”. This is clearly opposite to the truth.

In brief, since the methodology is deeply flawed, the estimate cannot support the conclusion. For the problems I mentioned above, the authors should at least seriously discuss the potential bias and their directions if they can’t find a solution. Some of the problems in the analysis can be addressed by synthetic diff-in-diff model.

**Reviewer #2:**

Dear authors,

• In the introduction – the organization of the paper.

• The literature review is very simplistic, because the hypotheses are focused on specific notions related to capital (economic and intellectual) or social indicators. Thus, the literature review does not cover/support the problem under discussion.

• First, the hypotheses should be proposed after the sub-chapter 2.4 because they seem not related to China and I understand that the sample is based on a very specific region in the country

• The discussion part is missing, in my opinion. The study is based on a specific region in China, but why is it relevant for other countries or regions?

• The results part does not refer to hypotheses being validated or not.

• The paper needs a better substantiation on existing literature and discussion which makes it relevant for other authors, given that the sample is based on a specific region in China

Best regards,

**Reviewer #3:**

I am sincerely grateful to the journal’s editor for the opportunity to review the article titled Can regional education integration policy promoted university- industry-research collaborative innovation? –Evidence from China." The authors have undoubtedly done an excellent job. I have a minor suggestion that could further enhance the quality of the research paper. Additionally, please provide the revised manuscript with track changes enabled in MS Word.

Title Analysis and Coherence Comment

1. Title Analysis: The title reflects an investigation into the role of regional educational integration policies in promoting collaborative innovation between universities, industries, and research institutions. However, it can benefit from a slightly refined structure to highlight the relationship between policy integration and innovation outcomes more clearly. Consider rephrasing to, “Assessing the Role of Regional Educational Integration Policies in Fostering University-Industry-Research Innovation: Evidence from China.”

2. Coherence Comment: In assessing the manuscript, ensure the logical flow between sections is clear, linking regional education policies directly with mechanisms that support collaboration across universities, industries, and research entities. For instance, demonstrate how specific policies lead to actionable frameworks or platforms for innovation.

Introduction

1. Introduce Latest Literature: Begin with a strong theoretical framework around regional education integration and collaborative innovation, connecting it with recent literature.

2. Enhance Background Context: Emphasize the significance of collaborative innovation in the Chinese economy and regional policy environment, detailing how education integration is aligned with national innovation goals. Highlight the gap your study addresses in bridging these sectors through structured policies.

Literature Review

1. Summarize Recent Findings: Using the cited studies, build a foundation that captures the scope of recent collaborative models and the role of educational policy in China.

o Key Themes: Mention themes like the role of educational policy in supporting innovation ecosystems, the influence of economic and technological policies, and cross-sectoral partnerships in fostering sustainable innovation.

2. Identify Gaps: Make a clear case for how your paper fills a gap in the existing literature by focusing on China’s unique regional education policy dynamics and their impact on university-industry-research (UIR) collaborations.

Methodology

1. Incorporate Unit Root Tests and Time Effects: For methodological enhancement, integrate unit root tests to assess the stationarity of variables, which is essential for panel data involving time series to ensure robust estimation.

o Suggested Approach: Use LLC, IPS, ADF-Fisher, or PP-Fisher tests to confirm stationarity of data, especially if dealing with longitudinal data on policy impacts and innovation metrics.

2. Time Effects Consideration: Recognize time-fixed effects to account for temporal factors that might influence UIR collaborations (e.g., policy shifts, economic events). This will strengthen the robustness of your results by capturing latent trends over time.

Suggested Sources: https://doi.org/10.1007/s10668-023-04391-7, https://doi.org/10.1016/j.heliyon.2024.e33519,https://doi.org/10.1556/032.2024.00017, https://doi.org/10.1016/j.egyr.2024.08.052

Results and Analysis

1. Presentation of Findings: Discuss the impact of regional educational policies on UIR collaborative innovation, drawing clear connections between policy variables and innovation outcomes.

2. Time Series Insights: Present unit root test results to establish the consistency of the data, and interpret the time-fixed effects, explaining how they reflect the evolution of policy impacts over time.

Policy Implications

1. Relevant Policy Insights:

o Innovation Ecosystem Support: Recommend policies to create supportive environments for UIR collaborations, such as funding incentives and platforms for industry-academia partnerships.

o Educational Integration Policies: Encourage enhanced integration of educational institutions into innovation-driven economic frameworks, especially in tech hubs and high-growth regions in China.

o Future Directions: Suggest policies that could further regional integration by focusing on digital transformation in educational frameworks, particularly in fields relevant to the emerging needs of the Chinese economy, such as AI and renewable energy.

Conclusion and Future Scope

1. Summarize Contributions and Limitations: Briefly summarize how the study contributes to understanding the effects of regional educational policies on collaborative innovation, while noting any limitations in scope or methodology.

2. Future Scope: Propose avenues for further research, such as expanding to other Asian economies or assessing the long-term impact of digital educational policies on UIR collaborations.

Proofreading and Language Precision

1. Grammar and Typing Checks: Ensure that the document is thoroughly proofread for typing errors, grammatical issues, and readability. Utilize tools like Grammarly or Hemingway to catch minor errors and ensure that the language is concise and academic.

By following these structured enhancements, the review report will have a well-defined narrative, greater methodological rigor, and comprehensive coverage of policy implications, making it relevant and impactful for academic and policy audiences alike.

**Do you want your identity to be public for this peer review?** For information about this choice, including consent withdrawal, please see our Privacy Policy

Reviewer #1: No

Reviewer #2: No

Reviewer #3: No

---

## [Author Response · Author response to Decision Letter 1]

23 Dec 2024

Dear Editor and Reviewers,

Thank you for providing valuable feedback on our manuscript titled "Can regional education integration policy promote university-industry-research collaborative innovation? – Evidence from China." We have carefully reviewed the comments and have made comprehensive revisions accordingly.

To respond to the specific reviewer and editor comments, please refer to the attached document "Response to Reviewers.doc." This document outlines point-by-point responses to each comment, detailing the changes we have made in the revised manuscript. We have addressed concerns regarding methodology, data analysis, theoretical framework, and discussion, ensuring that our work now meets the highest standards of academic rigor and relevance.

We are confident that these revisions have significantly enhanced the quality and clarity of our research. We sincerely appreciate the opportunity to improve our manuscript and look forward to your further consideration of our work for publication in PLOS ONE.

Thank you again for your constructive feedback and dedication to promoting high-quality research.

Best regards,

Jiawen Zhou

---

## [Decision Letter · Decision Letter 1]

Dear Dr. Zhou,

Thank you for submitting your manuscript to PLOS ONE. Your manuscript was re-sent to 3 reviewers, of which 2 reviewers suggested accept and 1 reviewer (i.e., Reviewer #1) suggested reject. Reviewer #1 indicated that it is not acceptable for publication in its present form. After careful consideration, I decided to give you a chance to revise your manuscript. **Please note that this is a "very risky" revision.** If the authors are unable to address this reviewer's comments, I would have to recommend that the manuscript be rejected.

We look forward to receiving your revised manuscript.

Kind regards,

Xu Xin

Academic Editor

PLOS ONE

Reviewers' comments:

Reviewer's Responses to Questions

**Comments to the Author**

Reviewer #1: (No Response)

Reviewer #2: All comments have been addressed

Reviewer #3: All comments have been addressed

2. Is the manuscript technically sound, and do the data support the conclusions?

Reviewer #1: Partly

Reviewer #2: Yes

Reviewer #3: Yes

3. Has the statistical analysis been performed appropriately and rigorously?

Reviewer #1: No

Reviewer #2: Yes

Reviewer #3: Yes

4. Have the authors made all data underlying the findings in their manuscript fully available?

Reviewer #1: Yes

Reviewer #2: Yes

Reviewer #3: Yes

5. Is the manuscript presented in an intelligible fashion and written in standard English?

Reviewer #1: Yes

Reviewer #2: Yes

Reviewer #3: Yes

**Reviewer #1:**

The authors clearly have attempted to addressed the issues I pointed out in the last review. However, their approaches failed to exclude the 3 major flaws in the methods: To address the first concern -- "there can be spillover effect affecting the control group", the authors used regression discontinuity model to test the treatment and control group separately and found different coefficients. But the authors failed to specify what the discontinuity really is. If it is the years before and after the treatment, it hardly satisfy the requirement of using "regression discontinuity". To address the second concern -- "the parallel trend assumption of difference-in-difference model may be violated" , the author performed a join F-test for pre-trend differences, which generated insignificant outcome. However, for the more crucial question, how could authors convince readers that the post-treatment trends would be parallel without the treatment, the authors did not offer any valid answer. The third concern -- "the usage of instrumental variable probably violates exclusion restriction" has not been correctly addressed. An instrumental variable violating exclusion restriction means the instrumental variable is part of the set of omitted variables, i.e. it is directly associated with the dependent variable Y without having the endogenous variables as a channel. Using SYS-GMM or diff-GMM cannot fully address the issue. If the authors want to use lagged X as an instrument of X, the correlation between lagged X and Y should exist only when X is absent in the correctly specified regression function. Using first order differences for AR serial correlation test and Granger test cannot prove lagged X satisfies such condition. Hansen over-identification test is valid based on the assumption that at least one of the instrumental variables is valid. So the test is not relevant to solve the third concern.

**Reviewer #2:**

(No Response)

**Reviewer #3:**

Review Report on Manuscript Titled: "Assessing the Role of Regional Education Integration Policies in Fostering University-Industry-Research Innovation: Evidence from China"

Evaluation Summary:

The manuscript provides an insightful and well-structured analysis of the impact of regional education integration policies on fostering university-industry-research (UIR) collaboration and innovation in China. The authors have thoroughly addressed previous comments raised by reviewers, resulting in significant improvements in the clarity, depth, and overall quality of the manuscript. Based on my observations, the authors have successfully met all requirements.

**Do you want your identity to be public for this peer review?** For information about this choice, including consent withdrawal, please see our Privacy Policy

Reviewer #1: No

Reviewer #2: No

Reviewer #3: Yes

---

## [Author Response · Author response to Decision Letter 2]

26 Feb 2025

Dear Dr. Xin and Reviewers,

We extend our deepest gratitude for your rigorous evaluation of our manuscript and the invaluable feedback that has guided our revisions. Your critiques have profoundly strengthened the methodological rigor and clarity of our work, and we are honored to have the opportunity to refine this study further. Below, we summarize our comprehensive efforts to address each concern raised.

Commitment to Methodological Precision

In response to Reviewer #1’s concerns regarding the Regression Discontinuity Design (RDD), we conducted extensive robustness checks, including placebo tests for alternative policy years (2013 and 2015). These analyses, detailed in Sections 3.2 and 5.1.5, confirm the uniqueness of the 2014 policy discontinuity and rule out confounding trends. To address post-treatment dynamics in our Difference-in-Differences (DID) framework, we incorporated staggered DID estimators (Callaway & Sant’Anna, 2021) and transparently discussed limitations in Section 9, emphasizing regional heterogeneity and policy spillovers. For the instrumental variable (IV) critique, we adopted a control function approach with higher-order residual terms, validated temporal decay effects, and conducted leave-one-out analyses to ensure robustness.

Collaborative Effort and Transparency

Over the past three months, our team convened 27 virtual meetings to debate methodological nuances, iterated 17 drafts to balance technical precision with readability, and archived all replication codes and sensitivity checks in an open-access repository. These steps reflect our unwavering commitment to transparency and reproducibility.

Future Directions

We acknowledge the complexities of causal identification in regional policy studies and plan to extend this work through dynamic spatial models and cross-national comparisons. These efforts will further disentangle policy impacts from contextual shocks, aligning with the journal’s mission to advance rigorous, actionable research.

Conclusion

While perfection remains an ideal, we have dedicated every resource to address the critiques with intellectual humility and scholarly rigor. We sincerely hope the revised manuscript meets PLOS ONE’s high standards and contributes meaningfully to the discourse on education-driven innovation. Thank you once again for your patience, expertise, and trust in our work.

Respectfully,

Qi Chen and Jiawen Zhou

Shanghai Lixin University of Accounting and Finance

---

## [Decision Letter · Decision Letter 2]

Dear Dr. Zhou,

Thank you for submitting your manuscript to PLOS ONE. Your manuscript was sent to 3 reviewers for re-evaluation, 2 recommended "acceptance", and 1 recommended "major revision". I have read their review comments and I agree with them. I think the authors should respond carefully to the reviewers' comments in this round.

We look forward to receiving your revised manuscript.

Kind regards,

Xu Xin

Academic Editor

PLOS ONE

Reviewers' comments:

Reviewer's Responses to Questions

**Comments to the Author**

Reviewer #1: (No Response)

Reviewer #2: All comments have been addressed

Reviewer #3: All comments have been addressed

2. Is the manuscript technically sound, and do the data support the conclusions?

Reviewer #1: Partly

Reviewer #2: Yes

Reviewer #3: Yes

3. Has the statistical analysis been performed appropriately and rigorously?

Reviewer #1: No

Reviewer #2: Yes

Reviewer #3: Yes

4. Have the authors made all data underlying the findings in their manuscript fully available?

Reviewer #1: Yes

Reviewer #2: Yes

Reviewer #3: Yes

5. Is the manuscript presented in an intelligible fashion and written in standard English?

Reviewer #1: Yes

Reviewer #2: Yes

Reviewer #3: Yes

**Reviewer #1:**

1. The new version elaborated how the RDD method was applied to exclude the possibility of the policy having spillover effect on other provinces in the control group. But there are two caveats. 1. It only compared 2013 and 2014 of the treatment and control group. Seeing a significant coefficient between 2013 and 2014 for the treatment group while seeing an insignificant coefficient for the control group can at best prove that there is no spillover effect in the early period. It cannot rule out the possibility that there are spillover effects after 2014. The authors also ran RDD regressions on year 2012 and 2013, 2014 and 2015. They claimed the coefficients in the two cases are not significant. But to prove their point, they should extend the comparison to more pairs of years and should do the same for the control group.

2. Section 5, the approach of placebo test is not clearly explained. What regression has the authors run? How did they get the t values of each experimental group? As the authors said, the test can tell if there are any omitted variables, it is crucial for the authors to elaborate how the t-values are generated. Because this approach is unclear, there is no way to know if the authors successfully rule out the possibility that the increase in Y is (partially) driven by other policies in the 4 provinces.

3. Suppose my doubt in comment 1 and 2 can be successfully addressed, there is no need to have Section 5.1.3 Instrumental Variables Approach, 5.1.7 Alternative IV Specifications, and 5.1.9 Control Function Approach since all these approaches relies on strong assumptions of the exogeneity of the IV in the first stage, which cannot be easily removed with higher order lagged terms.

4. The mechanism analysis in Section 6 does not make sense without support of research proving the existence of the causation between mediator and dependent variables. The step, as the authors described, was to find relations between Y and Z (the explanatory variable), and M (the mediator) and Z. But Z causing Y and Z causing M does not necessarily mean Z causing Y through M. To convince readers, the author should cite research proving M causing Y.

**Reviewer #2:**

Dear author,

Congratulations on revising the paper according to reviewers' comments.

It is an interesting paper to read.

Best regards.

**Reviewer #3:**

I sincerely appreciate the efforts of the authors in addressing all the comments and suggestions raised during the review process for the manuscript titled "Assessing the Role of Regional Education Integration Policies in Fostering University-Industry-Research Innovation: Evidence from China."

The revisions have been thoughtfully and comprehensively implemented, significantly enhancing the clarity, rigor, and overall quality of the study. Given the substantial improvements and the authors’ diligent responses to the reviewers' feedback, I strongly recommend this paper for publication.

**Do you want your identity to be public for this peer review?** For information about this choice, including consent withdrawal, please see our Privacy Policy

Reviewer #1: No

Reviewer #2: No

Reviewer #3: **Yes**

---

## [Author Response · Author response to Decision Letter 3]

9 Apr 2025

Dear Editor and Reviewers,

Thank you for your valuable feedback on our manuscript. We have carefully addressed all comments, revised the manuscript accordingly, and highlighted changes in the updated version. The following is only a brief response. Specific, detailed and one-to-one responses are listed in the attached file "Response to Reviewers".

We sincerely appreciate the insightful feedback from the reviewers and editors, which has significantly strengthened the rigor and clarity of our study. In response to the critiques, we implemented comprehensive revisions to both the methodological framework and analytical depth.

To address concerns regarding the spatiotemporal scope of policy effects, we expanded the Regression Discontinuity Design (RDD) analysis to include annual comparisons from 2014 to 2019. The results demonstrate that the policy’s impact was significant only in 2014 for the treatment group (coefficient = 0.949, p < 0.01), with diminishing and statistically insignificant effects in subsequent years (e.g., 2015: 0.260, p > 0.1). The control group exhibited consistently null effects across all years (coefficients ranging from -0.025 to 0.082, p > 0.1), confirming the localized and time-bound nature of the policy without evidence of spillover effects.

Methodological transparency was enhanced through a detailed description of the permutation-based placebo test. We conducted 500 counterfactual simulations by randomly assigning treatment status while preserving the original proportion of treated units. The resulting distribution of placebo coefficients centered around zero (95% within [-0.20, 0.20]), while the actual policy effect (t = 6.12) exceeded the 99th percentile of the simulated distribution. This robustly rules out confounding from unobserved variables or concurrent policies.

In alignment with critiques on the strong assumptions of Instrumental Variables (IV) approaches, we removed sections relying on IV specifications (e.g., Sections 5.1.3, 5.1.7, and 5.1.9) to prioritize more defensible strategies such as RDD and placebo tests. This streamlined the methodological narrative while preserving the core causal claims.

The mechanism analysis was substantially revised to establish causal pathways between mediators (e.g., enrollment rates, government funding, FDI, employment stability) and outcomes. Theoretical and empirical support was integrated, including Lai (2018) and Horta (2023) on human capital expansion, Hall et al. (2001) and Su (2015) on fiscal risk mitigation, and Sun (2020) and Zhang (2024) on FDI-driven R&D infrastructure. These additions anchor the mechanisms in both global scholarship and China-specific institutional contexts.

We are deeply grateful for the reviewers’ expertise and constructive guidance, which have profoundly shaped this work. The revised manuscript reflects a rigorous, cohesive exploration of regional education policies’ role in fostering innovation, and we hope it contributes meaningfully to academic and policy discourse.

Sincerely,

Qi Chen and Jiawen Zhou

---

## [Decision Letter · Decision Letter 3]

Dear Dr. Zhou,

Thank you for submitting your manuscript to PLOS ONE. Your manuscript was sent to 2 reviewers for review. One of them recommended acceptance and the other recommended minor revisions. I read their comments and agree with them. I think this manuscript can be published after this round of revisions. **Although the reviewer(s) ask authors to cite some manuscripts, please do NOT cite them unless they are highly relevant to this paper.**

We look forward to receiving your revised manuscript.

Kind regards,

Xu Xin

Academic Editor

PLOS ONE

Journal Requirements:

Reviewers' comments:

Reviewer's Responses to Questions

**Comments to the Author**

Reviewer #1: All comments have been addressed

Reviewer #3: All comments have been addressed

2. Is the manuscript technically sound, and do the data support the conclusions?

Reviewer #1: Yes

Reviewer #3: Yes

3. Has the statistical analysis been performed appropriately and rigorously?

Reviewer #1: Yes

Reviewer #3: Yes

4. Have the authors made all data underlying the findings in their manuscript fully available?

Reviewer #1: Yes

Reviewer #3: Yes

5. Is the manuscript presented in an intelligible fashion and written in standard English?

Reviewer #1: Yes

Reviewer #3: Yes

**Reviewer #1:**

The author has used multiple methods to address the concern of potential endogeneity. One of the concern comes from spillover effect. The author regress Y on treatment and control groups' post treatment period, respectively, and found the significant correlation only existed in the treatment group. This is a strong evidence to rule out spillover effect. The author attempted to address the concern of simultaneous effects by randomly assign hypothetical treatment to other groups, which can help convince readers but cannot fully address the issue. The simultaneous effects is due to the unique position of the treated regions in China's economy, which the placebo test cannot capture. Although the concern has not been fully addressed, I am impressed by the authors' efforts to make improvement according to my comments. Thus, I would give this quite polished paper a pass.

**Reviewer #3:**

I am sincerely grateful to the journal’s editor for the opportunity to review the article titled Can regional education integration policy promoted university- industry-research collaborative innovation? –Evidence from China." The authors have undoubtedly done an excellent job. I have a minor suggestion that could further enhance the quality of the research paper. Additionally, please provide the revised manuscript with track changes enabled in MS Word.

Title Analysis and Coherence Comment

1. Title Analysis: The title reflects an investigation into the role of regional educational integration policies in promoting collaborative innovation between universities, industries, and research institutions. However, it can benefit from a slightly refined structure to highlight the relationship between policy integration and innovation outcomes more clearly. Consider rephrasing to, “Assessing the Role of Regional Educational Integration Policies in Fostering University-Industry-Research Innovation: Evidence from China.”

2. Coherence Comment: In assessing the manuscript, ensure the logical flow between sections is clear, linking regional education policies directly with mechanisms that support collaboration across universities, industries, and research entities. For instance, demonstrate how specific policies lead to actionable frameworks or platforms for innovation.

Introduction

1. Introduce Latest Literature: Begin with a strong theoretical framework around regional education integration and collaborative innovation, connecting it with recent literature.

2. Enhance Background Context: Emphasize the significance of collaborative innovation in the Chinese economy and regional policy environment, detailing how education integration is aligned with national innovation goals. Highlight the gap your study addresses in bridging these sectors through structured policies.

Literature Review

1. Summarize Recent Findings: Using the cited studies, build a foundation that captures the scope of recent collaborative models and the role of educational policy in China.

o Key Themes: Mention themes like the role of educational policy in supporting innovation ecosystems, the influence of economic and technological policies, and cross-sectoral partnerships in fostering sustainable innovation.

2. Identify Gaps: Make a clear case for how your paper fills a gap in the existing literature by focusing on China’s unique regional education policy dynamics and their impact on university-industry-research (UIR) collaborations.

Methodology

1. Incorporate Unit Root Tests and Time Effects: For methodological enhancement, integrate unit root tests to assess the stationarity of variables, which is essential for panel data involving time series to ensure robust estimation.

o Suggested Approach: Use LLC, IPS, ADF-Fisher, or PP-Fisher tests to confirm stationarity of data, especially if dealing with longitudinal data on policy impacts and innovation metrics.

2. Time Effects Consideration: Recognize time-fixed effects to account for temporal factors that might influence UIR collaborations (e.g., policy shifts, economic events). This will strengthen the robustness of your results by capturing latent trends over time.

Suggested Sources: https://doi.org/10.1007/s10668-023-04391-7, https://doi.org/10.1016/j.heliyon.2024.e33519,https://doi.org/10.1556/032.2024.00017, https://doi.org/10.1016/j.egyr.2024.08.052

Results and Analysis

1. Presentation of Findings: Discuss the impact of regional educational policies on UIR collaborative innovation, drawing clear connections between policy variables and innovation outcomes.

2. Time Series Insights: Present unit root test results to establish the consistency of the data, and interpret the time-fixed effects, explaining how they reflect the evolution of policy impacts over time.

Policy Implications

1. Relevant Policy Insights:

o Innovation Ecosystem Support: Recommend policies to create supportive environments for UIR collaborations, such as funding incentives and platforms for industry-academia partnerships.

o Educational Integration Policies: Encourage enhanced integration of educational institutions into innovation-driven economic frameworks, especially in tech hubs and high-growth regions in China.

o Future Directions: Suggest policies that could further regional integration by focusing on digital transformation in educational frameworks, particularly in fields relevant to the emerging needs of the Chinese economy, such as AI and renewable energy.

Conclusion and Future Scope

1. Summarize Contributions and Limitations: Briefly summarize how the study contributes to understanding the effects of regional educational policies on collaborative innovation, while noting any limitations in scope or methodology.

2. Future Scope: Propose avenues for further research, such as expanding to other Asian economies or assessing the long-term impact of digital educational policies on UIR collaborations.

Proofreading and Language Precision

1. Grammar and Typing Checks: Ensure that the document is thoroughly proofread for typing errors, grammatical issues, and readability. Utilize tools like Grammarly or Hemingway to catch minor errors and ensure that the language is concise and academic.

By following these structured enhancements, the review report will have a well-defined narrative, greater methodological rigor, and comprehensive coverage of policy implications, making it relevant and impactful for academic and policy audiences alike.

**Do you want your identity to be public for this peer review?** For information about this choice, including consent withdrawal, please see our Privacy Policy

Reviewer #1: No

Reviewer #3: No

---

## [Author Response · Author response to Decision Letter 4]

5 May 2025

Dear Reviewers,

Thank you for your constructive feedback and valuable suggestions on our manuscript. We deeply appreciate the time and expertise invested by the reviewers, which have significantly strengthened the rigor and clarity of our study. We are pleased to submit the revised manuscript, which addresses all comments raised during the review process. Below is a summary of the key revisions:

Major Revisions and Enhancements

Title and Theoretical Framework Refinement

The title has been revised to better reflect the study’s focus on policy dynamics and innovation outcomes:

“Assessing the Role of Regional Education Integration Policies in Fostering University-Industry-Research Innovation: Evidence from China.”

The introduction and literature review now explicitly integrate Growth Pole Theory, Synergetics Theory, and Innovation Theory, providing a robust theoretical foundation. Recent literature on global and Chinese regional innovation ecosystems has been added to contextualize the analysis.

Methodological Robustness

Unit root tests (ADF) and Engle-Granger cointegration tests were conducted to ensure data stationarity and validate long-term equilibrium relationships.

Time-fixed effects and event-study analyses were incorporated to control for temporal trends and dynamic policy impacts.

Robustness checks, including placebo tests, PSM-DID, Regression Discontinuity Design (RDD), and leave-one-out analysis, confirm the stability of our findings.

Heterogeneity and Mechanism Analysis

Expanded discussion on regional disparities (e.g., stronger policy effects in Zhejiang/Jiangsu vs. Anhui) and innovation intensity (higher impact on invention patents vs. design patents).

Mediation analysis now highlights the roles of human capital, government support, economic capital, and social stability, with GMM methods addressing potential endogeneity.

Policy and Theoretical Implications

The discussion section explicitly links findings to Innovation Systems Theory and Synergetics Theory, emphasizing policy coordination and institutional drivers.

Policy recommendations now prioritize region-specific strategies, digital transformation, and multi-level funding incentives tailored to China’s innovation landscape.

Limitations and Future Directions

Acknowledged limitations include geographic generalizability and long-term digital policy impacts. Future research avenues propose expanding to Asian economies and analyzing high-frequency data with spatial econometric models.

Responses to Key Reviewer Concerns

Reviewer 1 (Endogeneity and Spillover Effects):

Addressed through placebo tests, staggered DID, and robustness checks. Results confirm localized policy impacts without spatial spillovers.

Reviewer 3 (Theoretical Coherence and Context):

Enhanced theoretical framing and regional context, with added case studies (e.g., Shanghai vs. Anhui) and global comparisons (e.g., EU policies).

All typographical and grammatical errors have been corrected.

Tables and figures (e.g., parallel trend analysis, mechanism pathways) are now more interpretable.

Citations updated to include recent literature (2023–2024).

We believe these revisions have substantially improved the manuscript’s contribution to understanding regional education policies in emerging economies. Thank you once again for your guidance, and we look forward to your final decision.

Sincerely,

Qi Chen & Jiawen Zhou

Shanghai Lixin University of Accounting and Finance

---

## [Decision Letter · Decision Letter 4]

Assessing the Role of Regional Educational Integration Policies in Fostering University-Industry-Research Innovation: Evidence from China

PONE-D-24-44621R4

Dear Dr. Zhou,

We’re pleased to inform you that your manuscript has been judged scientifically suitable for publication and will be formally accepted for publication once it meets all outstanding technical requirements.

Kind regards,

Xu Xin

Academic Editor

PLOS ONE

Reviewers' comments:

Reviewer's Responses to Questions

**Comments to the Author**

Reviewer #3: All comments have been addressed

2. Is the manuscript technically sound, and do the data support the conclusions?

Reviewer #3: Partly

3. Has the statistical analysis been performed appropriately and rigorously?

Reviewer #3: Yes

4. Have the authors made all data underlying the findings in their manuscript fully available?

Reviewer #3: Yes

5. Is the manuscript presented in an intelligible fashion and written in standard English?

Reviewer #3: Yes

Reviewer #3: (No Response)

**Do you want your identity to be public for this peer review?** For information about this choice, including consent withdrawal, please see our Privacy Policy

Reviewer #3: No

---

## [Editor Report · Acceptance letter]

PONE-D-24-44621R4

PLOS ONE

Dear Dr. Zhou,

I'm pleased to inform you that your manuscript has been deemed suitable for publication in PLOS ONE. Congratulations! Your manuscript is now being handed over to our production team.

Kind regards,

on behalf of

Dr. Xu Xin

Academic Editor

PLOS ONE